



# A relaxed eddy accumulation flask sampling system for $^{14}$C-based partitioning of fossil and non-fossil $CO_2$ fluxes

Ann-Kristin Kunz[1,2], Lars Borchardt[3], Andreas Christen[1], Julian Della Coletta[2], Markus Eritt[3], Xochilt Gutiérrez[3], Josh Hashemi[1,4], Rainer Hilland[1], Armin Jordan[3], Richard Kneissl[3], Virgile Legendre[3], Ingeborg Levin[2,†], Susanne Preunkert[2], Pascal Rubli[5], Stavros Stagakis[6], and Samuel Hammer[2]

[1]Chair of Environmental Meteorology, Faculty of Environment and Natural Resources, University of Freiburg, Freiburg, Germany
[2]Institute of Environmental Physics, Heidelberg University, Heidelberg, Germany
[3]ICOS Flask and Calibration Laboratory, Max Planck Institute for Biogeochemistry, Jena, Germany
[4]Alfred Wegener Institute, Helmholtz Centre for Polar and Marine Research, Potsdam, Germany
[5]Empa, Materials, Science and Technology, Dübendorf, Switzerland
[6]Department of Environmental Sciences, University of Basel, Basel, Switzerland
[†]deceased, 10 February 2024

**Correspondence:** Ann-Kristin Kunz (ann-kristin.kunz@iup.uni-heidelberg.de)

**Abstract.**

A relaxed eddy accumulation (REA) system was developed and tested that enables conditional sampling of air for subsequent $^{14}CO_2$ analysis. It allows an observation-based partitioning of total $CO_2$ fluxes measured in urban environments by eddy covariance into fossil and non-fossil components. The purpose of this article is to describe the REA system, evaluate its performance and assess uncertainties. In the REA system, two separate inlet lines equipped with fast-response valves and loop systems adapted to the technical requirements enable the conditional collection of air in two sets of aluminum cylinders for updraft and downdraft samples, respectively. The switching between updraft sampling, downdraft sampling and stand-by mode is thereby determined by the vertical wind measured at 20 Hz by a co-located ultrasonic 3D anemometer. A logger program provides different options for the definition of a deadband, which is used to increase the concentration differences between updraft and downdraft samples. After the sampling interval, the accumulated air is transferred by an automated 24-port flask sampler into 3 l glass flasks, which can be analyzed in the laboratory, and the cylinders are re-evacuated for the next sampling. The REA system was tested in the laboratory as well as on a tall-tower near the city center of Zurich, Switzerland. Between July 2022 and April 2023, 103 REA up- and downdraft flask pairs and nine flask pairs from quality control tests were selected from the tall-tower for laboratory analysis based on suitable micro-meteorological conditions. Uncertainties in the $CO_2$ concentration differences between updraft and downdraft flasks were estimated by simulations using 20 Hz in situ measurements of a closed-path and an open-path gas analyzer co-located with the ultrasonic anemometer. The measurements show that there is no significant bias in the concentration differences between updraft and downdraft samples, and that uncertainties due to the sampling process are negligible when estimating fossil fuel $CO_2$ signals. In the Zurich measurements, the $CO_2$ concentration differences between the flask pairs agreed with the differences of in situ measurements within $-0.005 \pm 0.227$ ppm. The



largest source of uncertainty and main limitation in the separation of fossil and non-fossil $CO_2$ signals in Zurich was the small signal-to-noise ratio of the $\Delta^{14}C$ differences measured by accelerator mass spectrometry between the updraft and downdraft flasks. The novel REA flask sampling system meets the high technical requirements of the REA method and is a promising technology for observation-based estimation of fossil fuel $CO_2$ fluxes.

## 1 Introduction

In view of the overarching aim of reducing anthropogenic greenhouse gas emissions to mitigate climate change, reliable emissions data and timely information on emission reductions are indispensable, especially at the local scale in urban environments where emission reduction efforts are to be assessed. Of central importance in this context is the quantification of fossil fuel $CO_2$ (ff$CO_2$) emissions in cities, as cities contribute more than 70 % to global and European ff$CO_2$ emissions (Seto et al., 2014). Officially reported bottom-up emission inventories are usually based on statistical activity data, e.g., fossil fuel consumption,

and emission factors for the different emission sectors, such as traffic or industry (Super et al., 2020). Downscaled to urban and local resolutions, they form an important basis for policy decisions as well as fundamental research (WMO, 2022). Despite continuous improvements to such inventories, the benefits of bottom-up estimates is currently limited by their coarse spatial and temporal resolution, large uncertainties in the available methodologies, and the delayed availability of data (Gately and Hutyra, 2017; Lauvaux et al., 2020; Stagakis et al., 2023). To independently validate and refine emission inventories for $CO_2$,

atmospheric measurements providing timely, localized, and sector-specific top-down information are therefore indispensable.

The only method that allows direct measurement of vertical atmospheric trace gas fluxes is the eddy covariance (EC) technique, in which vertical wind velocity and trace gas concentrations are both measured at a frequency of 10 to 20 Hz (e.g., Mauder et al., 2021). Although this method assumes a horizontally flat and homogeneous surface area (Foken et al., 2012), studies have shown that EC measurements can also be successfully performed in a complex and heterogeneous urban envi-

ronment (Grimmond and Christen, 2012; Feigenwinter et al., 2012; Christen, 2014). However, EC-based $CO_2$ flux estimates contain fossil *and* non-fossil components. Models as well as measurements have shown that biospheric $CO_2$ fluxes (autotrophic and heterotrophic respiration and photosynthesis) can contribute significantly to the total $CO_2$ flux measured over an urban area, even in winter (e.g., Crawford et al., 2011; Kellett et al., 2013; Hardiman et al., 2017; Wu et al., 2022). In addition, human respiration fluxes can account for more than 10 % of the total annual $CO_2$ flux depending on the population density (Moriwaki

and Kanda, 2004; Kellett et al., 2013). Considering also the increasing use of biofuels (e.g., Guo et al., 2015), this results in significant uncertainties in inverse estimates of fossil fuel emissions (Crawford and Christen, 2015; Wu et al., 2018; Stagakis et al., 2023).

To separate fossil and non-fossil $CO_2$ enhancements, radiocarbon has proven to be the ideal tracer, since $^{14}C$-free fossil fuels dilute the atmospheric $^{14}C/C$ ratio compared to a clean background (e.g., Levin et al., 2003; Turnbull et al., 2016). However,

the lack of fast-response gas analyzers for $^{14}C$ prevents direct ff$CO_2$ flux measurements using the EC method (Rinne et al., 2021). The relaxed eddy accumulation (REA) method (Businger and Oncley, 1990), on the other hand, allows flux estimation from the concentration differences between two conditionally collected air samples, which can be determined in the laboratory.



The ffCO$_2$ fluxes can thus be estimated from the $^{14}$C-based ffCO$_2$ concentration differences between the sample pairs. To date, the most precise $^{14}$C measurements for small samples are based on accelerator mass spectrometry. For this purpose, the

CO$_2$ is extracted from the air and catalytically reduced to graphite (Lux, 2018).

In the present study, to our knowledge, the first REA system for $^{14}$C-based estimation of ffCO$_2$ fluxes was developed. Based on the principles of relaxed eddy accumulation (Sect. 2) and $^{14}$C-based fossil fuel CO$_2$ estimation (Sect. 3), the purpose of this article is to describe the novel REA flask sampling system (Sect. 4), evaluate its performance and assess the measurement uncertainties (Sect. 5). As a proof of concept of $^{14}$C-based separation of fossil and non-fossil CO$_2$ components, the ffCO$_2$

concentration differences between updraft and downdraft flasks collected during a field campaign on a tall-tower 112 m above ground level near the city center of Zurich, Switzerland, are presented (Sect. 6). This work forms the basis for the derivation and analysis of ffCO$_2$ fluxes in Zurich and for future deployments in other urban environments.

## 2    The relaxed eddy accumulation method

Relaxed eddy accumulation (REA), first described by Businger and Oncley (1990), is a conditional sampling method for

measuring turbulent trace gas fluxes using slow response analyzers. A fast ultrasonic anemometer measures the vertical wind velocity $w$ at a frequency of 10 to 20 Hz. Based on $w$, the opening and closing of two fast-response sampling valves is controlled in quasi realtime. Any bias in the vertical wind velocity must therefore be removed before activating the valves (Rinne et al., 2021). When there is an updraft eddy and $w$ is above a certain threshold $w_0$, air is collected and accumulated in an updraft reservoir, whereas air is collected in a separate downdraft reservoir when $w < -w_0$.

The range of wind speeds $[-w_0 : w_0]$ where no air is collected is called the deadband. Under ideal conditions, the mean vertical wind speed $\overline{w}$ over a sampling period of, for example, 30 or 60 min is zero and defines the center of the deadband. The deadband width can be constant or dynamically adjusted to the standard deviation of the vertical wind speed $\sigma_w$, so that $w_0 = \overline{w} + \delta\sigma_w$ (Rinne et al., 2021). The larger $\delta$, the greater the concentration difference between the updraft and downdraft reservoirs, reducing the relative measurement uncertainty (Rinne et al., 2021). In addition, a larger deadband reduces the

switching frequency of the sampling valves, thereby increasing their lifetime (Rinne et al., 2021). However, it also reduces the fraction of time during which air is collected, which reduces the sample volume and the statistical significance. A compromise between a high concentration difference, a sufficient sample volume, and good representativity has to be found (Christen et al., 2006). Since the requirement $\overline{w} = 0$ can be violated, in particular in urban environments with complex airflow, and the actual value of $\overline{w}$ is not known before the end of the sampling period, two options to trigger the valves based on vertical wind were

implemented in this study (see Sect. 4.1).

The flow rate into the two reservoirs is always zero when the valves are closed and constant when they are open. This relaxes the high technical requirements of the true eddy accumulation method proposed by Desjardins (1972), where the sampling rate is adjusted in quasi realtime to the magnitude of the vertical wind velocity (for a detailed description see Siebicke and Emad (2019) and Emad and Siebicke (2023)). However, the constant flow rate prevents a direct assessment of the mean vertical

turbulent flux of a trace gas $F_c$. With REA, $F_c$ is calculated according to Eq. (1) from the mean concentration difference



between the updraft and the downdraft sample $\Delta c$, the standard deviation of the vertical wind speed $\sigma_w$, the mean molar air density $\overline{\rho_m}$ and an empirical coefficient $\beta$:

$$F_c = \beta \sigma_w \overline{\rho_m}(c^\uparrow - c^\downarrow) = \beta \sigma_w \overline{\rho_m} \Delta c. \tag{1}$$

The proportionality factor $\beta$ depends on the joint probability distribution of variations of the vertical wind velocity and the trace gas concentration (Milne et al., 1999). Without a deadband and under an ideal joint Gaussian distribution of $w$ and $c$, $\beta$ is 0.627 (Wyngaard, 1992; Baker et al., 1992), but the dependence of $\beta$ on the prevailing atmospheric conditions, the deadband width, and eventually on the scalar, e.g., the trace gas concentration, itself adds uncertainty to the calculated flux (Siebicke and Emad, 2019). Grönholm et al. (2008) showed that with a dynamic deadband, $\beta$ does not depend on the atmospheric stability, allowing the use of a constant value in flux calculations. Consequently, assuming similarity in the turbulent exchange of two quantities, $\beta$ can be determined with Eq. (1) from a scalar where the flux is known from EC measurements, e.g., temperature or $CO_2$ (Rinne et al., 2021).

Estimating the $ffCO_2$ difference between updraft and downdraft reservoirs based on $^{14}CO_2$ analyses of REA sample pairs and assuming scalar similarity between $CO_2$ and $^{14}CO_2$, Eq. (1) allows to determine the fossil contribution to a total $CO_2$ flux:

$$\frac{F_{ffCO_2}}{F_{CO_2}} = \frac{\Delta ffCO_2}{\Delta CO_2}. \tag{2}$$

Multiplying the $\Delta ffCO_2/\Delta CO_2$ ratio of REA samples with $F_{CO_2}$ obtained from EC measurements thus provides an observation-based $ffCO_2$ flux estimate.

It should be noted that Eq. (2) is unstable for $F_{CO_2}$ and/or $\Delta CO_2$ close to zero. In this case, a proxy other than $CO_2$ is needed, e.g., CO. Moreover, like any other method based on the eddy covariance principle, REA only provides reasonable estimates of the mean vertical turbulent flux if the micro-meteorological conditions are stationary and turbulence is well developed during the sampling period (Foken et al., 2012). If a temporal change in concentrations below the measurement height causes a significant storage flux, the measured turbulent fluxes are not representative of the respective surface fluxes at the time (Crawford and Christen, 2014; Bjorkegren et al., 2015). Further, if vertical advection leads to significant flux components due to non-turbulent vertical transport, this cannot be properly captured by eddy covariance and the REA approach (Foken et al., 2012). This implies the consideration of various criteria when selecting suitable samples in any application (see Sect. 4.4).

## 3  $^{14}C$-based fossil fuel $CO_2$ estimation

To determine the contribution of fossil fuel emissions to a measured $CO_2$ signal, $^{14}C$ has turned out to be an important tracer. The radioactive carbon isotope has a half-life of 5730 years. Consequently, the millions of years old fossil fuels are $^{14}C$-free, hence $CO_2$ emitted by fossil fuel combustion ($ffCO_2$) dilutes the atmospheric $^{14}C/C$ ratio. Due to the low $^{14}C$ abundance, the specific activity of a reservoir or sample is commonly given as relative deviation (in ‰) from the absolute specific activity of



the radiocarbon standard $A_{\text{ABS}} = 0.2261 \text{Bq gC}^{-1}$ (Stuiver and Polach, 1977):

$$\Delta^{14}\text{C} = \left( \frac{A_{\text{SN}}}{A_{\text{ABS}}} - 1 \right) \times 1000. \tag{3}$$

$A_{\text{SN}}$ is the $^{14}\text{C}$ sample activity normalized to the postulated mean $\delta^{13}\text{C}$-value of terrestrial wood of -25 ‰ to correct for isotopic fractionation due to biological or physical processes during the sample formation and the measurement routine (Stuiver and Polach, 1977).

Following the general approach of Turnbull et al. (2016) and Maier et al. (2023), any measured atmospheric $^{14}\text{CO}_2$ signal can, analogous to the total $CO_2$ concentration $c_{\text{meas}}$, be expressed as the sum of a background (bg), a fossil fuel (ff), a biofuel (bf), a nuclear (nuc), a stratospheric (strato), a respiratory (resp), a photosynthetic (photo), and an oceanic (oc) component:

$$c_{\text{meas}} = \sum_i c_i \tag{4}$$

$$c_{\text{meas}}\,^{14}\Delta = \sum_i c_i\,^{14}\Delta_i. \tag{5}$$

Here, $\Delta^{14}\text{C}$ has been abbreviated by $^{14}\Delta$ and $i$ = bg, ff, bf, nuc, strato, resp, photo, oc.

On a local scale, as during REA sampling, it can be assumed that two air parcels differ only in their fossil fuel, biofuel, respiration and photosynthesis components, while the impact of the other, more distant sinks and sources is the same. Since biofuels and respiration have a similar $^{14}\Delta$ signature, they cannot be distinguished by radiocarbon analysis alone, and are therefore summarized in the following as non-fossil (nf) $CO_2$ sources, with respiration being by far the largest contributor. Thus, the concentration differences between the updraft (↑) and the downdraft (↓) reservoir of a REA measurement can be expressed in the following way:

$$c^{\uparrow}_{\text{meas}} - c^{\downarrow}_{\text{meas}} = (c^{\uparrow}_{\text{ff}} - c^{\downarrow}_{\text{ff}}) + (c^{\uparrow}_{\text{nf}} - c^{\downarrow}_{\text{nf}}) + (c^{\uparrow}_{\text{photo}} - c^{\downarrow}_{\text{photo}}) \tag{6}$$

$$c^{\uparrow}_{\text{meas}}\,^{14}\Delta^{\uparrow}_{\text{meas}} - c^{\downarrow}_{\text{meas}}\,^{14}\Delta^{\downarrow}_{\text{meas}} = (c^{\uparrow}_{\text{ff}} - c^{\downarrow}_{\text{ff}}) \cdot^{14}\Delta_{\text{ff}} + (c^{\uparrow}_{\text{nf}} - c^{\downarrow}_{\text{nf}}) \cdot^{14}\Delta_{\text{nf}} + (c^{\uparrow}_{\text{photo}} - c^{\downarrow}_{\text{photo}}) \cdot^{14}\Delta_{\text{photo}}. \tag{7}$$

Combining Eq. (6) and Eq. (7), the difference in $c_{\text{ff}}$ between up and down reservoir can be estimated via:

$$c^{\uparrow}_{\text{ff}} - c^{\downarrow}_{\text{ff}} = \frac{1}{^{14}\Delta_{\text{photo}} -^{14}\Delta_{\text{ff}}} \left[ c^{\uparrow}_{\text{meas}}(^{14}\Delta_{\text{photo}} -^{14}\Delta^{\uparrow}_{\text{meas}}) - c^{\downarrow}_{\text{meas}}(^{14}\Delta_{\text{photo}} -^{14}\Delta^{\downarrow}_{\text{meas}}) + (c^{\uparrow}_{\text{nf}} - c^{\downarrow}_{\text{nf}})(^{14}\Delta_{\text{nf}} -^{14}\Delta_{\text{photo}}) \right]. \tag{8}$$

The corresponding uncertainty is derived according to Gauss' law of error propagation.

Since fossil fuel emissions do not contain any $^{14}\text{C}$, $^{14}\Delta_{\text{ff}}$ is per definition exactly -1000 ‰. On the contrary, biogenic signatures are heterogeneous and much more uncertain (e.g., Naegler and Levin, 2009b; Maier et al., 2023). To estimate $c^{\uparrow}_{\text{ff}} - c^{\downarrow}_{\text{ff}}$ (in the following denoted as $\Delta\text{ffCO}_2$), it is therefore necessary to make assumptions about $^{14}\Delta_{\text{photo}}$, $^{14}\Delta_{\text{nf}}$ and $c^{\uparrow}_{\text{nf}} - c^{\downarrow}_{\text{nf}}$.

For sites where biogenic and fossil sources are mixed and where the biogenic signal is dominated by the local biosphere, as is the case in cities, $^{14}\Delta_{\text{photo}}$ is best approximated by the current atmospheric signature $^{14}\Delta_{\text{meas}}$ (Maier et al., 2023). As there are always two flasks per REA run, the average of the up and down flasks is chosen: $^{14}\Delta_{\text{photo}} \approx 0.5 \cdot (^{14}\Delta^{\uparrow}_{\text{meas}} +^{14}\Delta^{\downarrow}_{\text{meas}})$. However, due to temporal variability, the $^{14}\Delta_{\text{photo}}$ uncertainty is larger than the measurement uncertainty and set to 10 ‰.





Non-fossil $CO_2$ (from autotrophic and heterotrophic respiration and to a lesser extent biofuels) is generally more enriched in $^{14}CO_2$, because heterotrophically respired $CO_2$ and $CO_2$ from biofuels was taken up by the biosphere several years to decades ago. At that time, the atmospheric $^{14}\Delta$ was higher due to $^{14}CO_2$ released during nuclear bomb tests in the 1950s/1960s, which was the dominant contribution up to the 2000's (Naegler and Levin, 2009a). Since then, the strongest component of the ongoing atmospheric $^{14}\Delta$ decline has been the emission of fossil fuel $CO_2$ (Levin et al., 2010; Turnbull et al., 2016). In total, the $^{14}\Delta$

signature of respiration was found to be larger than atmospheric values by a few tens of permil (e.g., Palonen et al., 2018; Chanca, 2022). Following Maier et al. (2023), an enrichment of the respiration-dominated $\Delta_{nf}$ of $25 \pm 12\ ‰$ is assumed in this study. The atmospheric signature during $CO_2$ uptake of the biosphere $\overline{\Delta_{atmo}}$ is estimated by the mean $^{14}\Delta_{meas}$ value in summer, when photosynthesis is most pronounced.

    The third unknown is the difference $\Delta c_{nf} = c_{nf}^{\uparrow} - c_{nf}^{\downarrow}$. It can be estimated based on the mean measured total $CO_2$ differences

between up and down flasks and the assumption that in an urban setting, there is always a fossil contribution. Since this is only a very rough and upper estimate, a relative uncertainty of $100\ \%$ is reasonable.

    The assumptions on $^{14}\Delta_{photo}$, $^{14}\Delta_{nf}$ and $\Delta c_{nf}$ as well as the specific values used to calculate $\Delta ffCO_2$ from the REA flasks of the first measurement campaign in Zurich, are summarized in Table 1.

**Table 1.** Variables used to estimate $c_{ff}^{\uparrow} - c_{ff}^{\downarrow} = \Delta ffCO_2$. $^{14}\Delta_i$ denote the $\Delta^{14}C$ values of fossil fuels (ff), photosynthetic (photo) and non-fossil (nf) $CO_2$, and flask measurements (meas). $\overline{^{14}\Delta_{meas}} = 0.5 \cdot (^{14}\Delta_{meas}^{\uparrow} + ^{14}\Delta_{meas}^{\downarrow})$ denotes the mean of the updraft and downdraft samples, which is different for each REA run. The atmospheric signature during $CO_2$ uptake of the biosphere $\overline{\Delta_{atmo}}$ is estimated by the mean $^{14}\Delta_{meas}$ value in summer (July to September 2022 in the case of the Zurich campaign). Also given are the specific values derived for the first measurement campaign in Zurich.

| Variable | Unit | Approximation | Zurich value |
|---|---|---|---|
| $^{14}\Delta_{ff}$ | $‰$ | $-1000$ | $-1000$ |
| $^{14}\Delta_{photo}$ | $‰$ | $\overline{^{14}\Delta_{meas}}$ | $\overline{^{14}\Delta_{meas}} \pm 10$ |
| $^{14}\Delta_{nf}$ | $‰$ | $\overline{^{14}\Delta_{atmo}} + 25$ | $9 \pm 16$ |
| $c_{nf}^{\uparrow} - c_{nf}^{\downarrow}$ | ppm | $\sim \overline{\Delta CO_2}$ | $5 \pm 5$ |

    Although $^{14}\Delta_{photo}$, $^{14}\Delta_{nf}$ and $\Delta c_{nf}$ are not well known, the uncertainty of the $\Delta ffCO_2$ estimates is dominated by the $^{14}CO_2$

measurement uncertainty, which currently leads to an inherent $\Delta ffCO_2$ uncertainty of at least $0.7\ ppm$, on average $1.2\ ppm$. Details are given in Appendix D.

## 4   Setup of the REA system

The relaxed eddy accumulation (REA) flask sampling system consists of an eddy covariance (EC) system with a 3D ultrasonic anemometer and an integrated open-path gas analyzer (IRGASON, Campbell Scientific, Inc., Logan, UT, USA), two fast-

response valves controlled by a solid state DC control module (SDM-CD8S, Campbell Scientific, Inc., Logan, UT, USA), a



data logger (CR6, Campbell Scientific, Inc., Logan, UT, USA) and an extension of a regular 24-port ICOS (Integrated Carbon Observation System) automated flask sampler, described for example in Levin et al. (2020) (Fig. 1). It allows the collection of updraft and downdraft air samples (Sect. 2) in ICOS glass flasks for subsequent laboratory analysis.

As depicted in Fig. 1, there are two inlets, one for updraft and one for downdraft conditions, about 20 cm away from the center of the ultrasonic anemometer. The collection of air is controlled by the two solenoid valves located approximately 30 cm behind the inlets. They respond to the data logger's 20 Hz signal, which is based on the vertical wind velocity measurements of the IRGASON (Sect. 4.1). This signal is also sent to the sampler computer, which controls the extended flask sampler. The sampled air is pumped into two separate 50 l cylinders, which are called buffers in the following. Two so-called loop systems (Sect. 4.2) avoid flow rate fluctuations during sampling and non-sampling to ensure constant flow rates despite high-frequency switching. Excess air from the updraft and downdraft sides is released through one common outflow. After a successful sampling event, for which both 50 l buffers need to have a pressure between 1.2 and 1.6 bar (Sect. 4.3), the samples are dried and transferred into 3 l glass flasks and the buffers are evacuated again. Since this takes about 45 min, there is a second pair of buffers that can be filled meanwhile. This allows a nearly continuous sampling routine. The flask pairs can be sent to the laboratory or re-sampled if, for example, the sampling conditions did not fulfill the requirements (Sect. 4.4). In addition, a third line from the

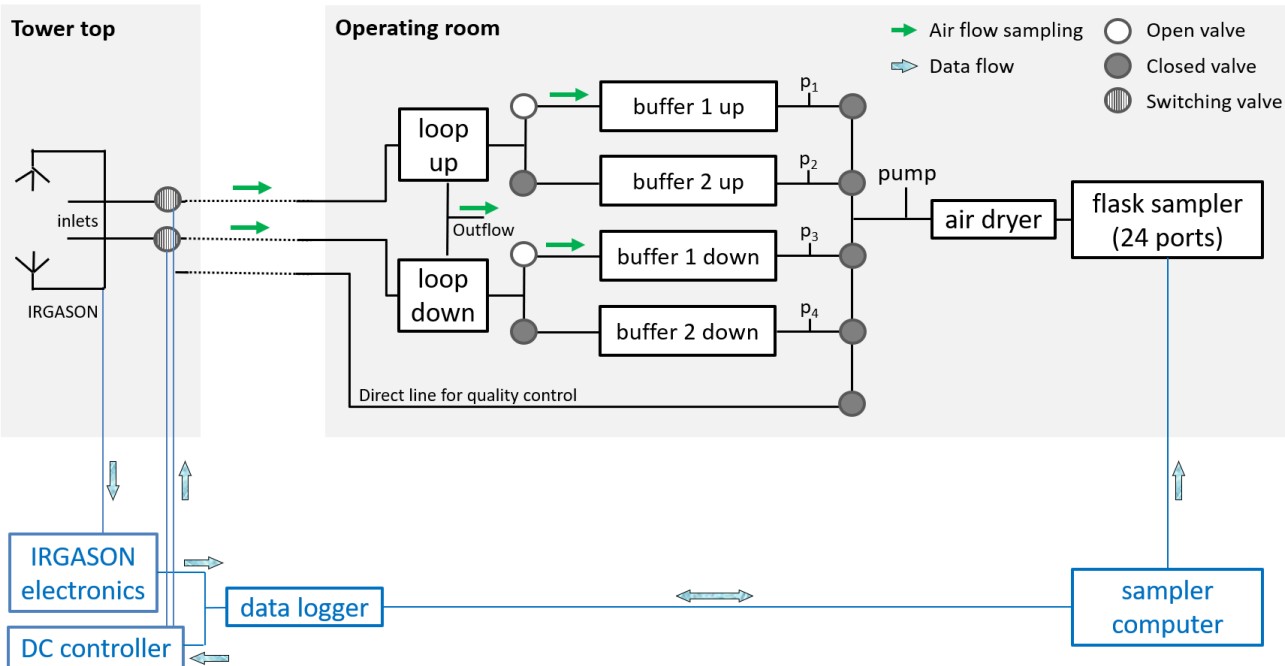

**Figure 1.** Schematic setup of the REA flask sampling system. Blue components at the bottom of the diagram depict the general wiring for data transfer and communication between the data logger and the sampler computer. The green arrows and the filled / unfilled / hatched circles indicate the air flow and the position of the valves when sampling into buffer set 1. Details of the loop systems are shown in Fig. 2.



tower top directly to the flask sampler enables a simultaneous sampling of flasks with a regular $1/t$ filling (Levin et al., 2020), which can be used for quality control tests (Sect. 5.3.1). In the following, the individual steps of collecting REA flasks are described in more detail. The following terminology is used: REA run: REA sampling event of usually 1 h; REA ID: unique, consecutively assigned number for each REA run; sampling mode: state of the up- / downdraft line when air is collected, i.e, the vertical wind is above / below the deadband; standby mode: state of the up- / downdraft line when no air is collected, i.e,

when the vertical wind is below / above or within the deadband. A photo and specifications of the components of the system are given in Appendix A.

## 4.1 Conditional collection of air

During a REA run, the data logger controls the opening and closing of the solenoid valves at the inlets according to the in situ measurements of the vertical wind velocity. Depending on the definition of the deadband (Sect. 2), flags are assigned to each

20 Hz wind measurement, denoting the status of the two valves in the current and previous time step. If the two values are different, the corresponding valve is switched. For quality control, the flags, in the following called REA flags, can be used to estimate the sample $CO_2$ concentrations from high-frequency in situ concentration measurements (see Sect. 5.2 and Sect. 5.3.2).

Table 2 shows the possible deadband settings implemented in the logger program, which can be selected depending on the

scientific question and site-specific requirements.

**Table 2.** Variables in the logger program that determine the deadband of the REA run.

| Variable | Values | Description |
|---|---|---|
| REA_DeadBandWidth | $\geq 0$ | Deadband width $\delta$ that, multiplied by the standard deviation of the vertical wind $\sigma_w$, determines the width of the deadband |
| REA_FreezeStatistics | 0 | Dynamic wind statistics: Mean and standard deviation of the vertical wind velocity $\overline{w}$ and $\sigma_w$ are calculated from a backward-looking moving average window. |
|  | 1 | Pre-set wind statistics: $\overline{w}$ and $\sigma_w$ are calculated from a certain time period before the sampling start. |
| SYS_MovingBlockDuration | $> 0$ | Length of the backward averaging interval in seconds used to calculate $\overline{w}$ and $\sigma_w$ (for both dynamic and pre-set wind statistics). |

It should be noted that the digital output from the IRGASON is delayed by several hundred milliseconds, depending on the bandwidth of the low-pass filter that is applied to the actual 60 Hz measurements. The larger the bandwidth, the smaller the





delay, with a minimum delay of 200 ms (20 Hz bandwidth). In addition there will also be a short delay between the signals being sent, the valves being physically switched and the air being sucked in due to the slight underpressure in the line. The

impact of these delays on the flask concentration differences is analyzed in Sect. 5.2.1.

## 4.2   Transfer of collected air into buffers

To pump the collected air at a constant flow rate into the respective buffers while ensuring that the system can switch between sampling and standby mode at any time, loop systems (Fig. 2) were developed. They consist of a membrane pump that runs continuously, a pressure control valve, a mass flow controller (MFC) and two three-way valves. Technical details of the

components are given in Appendix A.

When the solenoid valve at the inlet is closed (standby mode, i.e., when no air is sampled), the three-way valves are closed to the outside and the air circulates in the loop as shown by the red arrows. In sampling mode, both three-way valves switch to the normally closed (NC) position. Air is pumped through the line and the MFC into the buffer, while excess air leaves the system through the pressure control valve, as indicated by the green arrows. In both modes, the pump is continuously running

and the pressure behind the pump as well as the flow rate through the loop are constant due to the pressure control valve and the MFC. Thus, in sampling mode, the flow rate into the buffers is approximately constant and the system is able to switch between sampling and standby mode at any time. The effect of a remaining variability in the flow rates is discussed in Sect. 5.2.3.

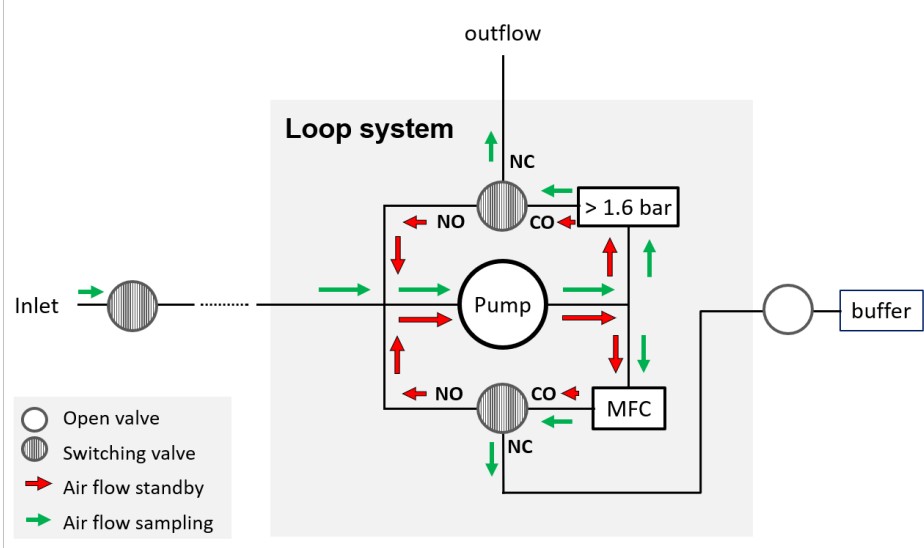

**Figure 2.** Schematic setup of the loop systems indicating the air flow in sampling and standby mode. NC: normally closed (closed in standby mode, open in sampling mode); NO: normally open (open in standby mode, closed in sampling mode); CO: constantly open; MFC: mass flow controller; > 1.6 bar: pressure control valve.





Each loop system has an internal volume $V_l = 12.0 \pm 1.1$ ml, while the volumes of the inlet tubes with length $l_t$ and radius
$r_t$ are $V_t = \pi r_t^2 l_t$. The pump flow velocity $q_{\text{pump}}$ [l min$^{-1}$] depends on the dimensions of the intake lines (longer and thinner tubes have a higher resistance) and the flow rate set at the MFC. The latter is adjusted based on the length of the REA run and the deadband width $\delta$: The shorter the REA run and the larger the deadband width, the larger the required flow rate into the buffers. Consequently, an air parcel needs the time

$$t_r = (V_l + V_t)/q_{\text{pump}} \tag{9}$$

from the inlet to the buffer. To avoid collecting air that was in the lines before the sampling start while the last part of the desired sample air is lost, a so-called rinse time of length $t_r$ was defined for the beginning and after the end of a sampling period. During the first $t_r$ seconds in sampling mode, the valves at the buffers remain closed so that the residual air is released through the outflow. On the contrary, the pump stays on and all valves remain open for an additional $t_r$ seconds after the end of the sampling period so that all sample air remaining in the tubes is transferred into the buffers. Due to the site-specific
lengths of the intake lines, $t_r$ must be individually calculated and adjusted in the sampler software. As shown in Sect. 5.1 and Sect. 5.2.2, calculated values agree well with measurements of the travel time of a $CO_2$ pulse and uncertainties in the flask concentrations resulting from the uncertainty of $t_r$ are negligible.

### 4.3 Transfer of accumulated air into ICOS glass flasks

After the sampling period, the accumulated air is transferred from the buffers into 3 l glass flasks in the flask sampler. Thereby,
it is dried to a dew point of approximately $-40\,^\circ$C. To (almost) completely replace the initial air content of a flask with the sample air from the buffers, the flask volume is flushed 10 times at atmospheric pressure. Then the flask is filled up to 2 bar. As the performance of the sampler transfer pump does not allow to use the entire sample volume in the buffers, a minimal sample amount of 60 l is desired, corresponding to a pressure of 1.2 bar in the buffers. Excess air is discarded when the buffers are evacuated. At the same time, the maximum pressure at which the pumps in the loop systems can be operated is 1.6 bar,
so a REA run is automatically stopped when either buffer reaches this threshold. Consequently, a REA sampling event is only successful if the pressures of updraft and downdraft buffer are between 1.2 and 1.6 bar at the end of sampling.

### 4.4 Sample selection for laboratory analysis

Within the ICOS network, flasks are analyzed with gas chromatography for $CO_2$, CO, $CH_4$, $N_2O$, $SF_6$ and $H_2$ at the flask and calibration laboratory (FCL) in Jena, Germany. To measure $^{14}CO_2$, the flasks are sent to the ICOS Central Radiocarbon
Laboratory (CRL) in Heidelberg. There, the $CO_2$ is extracted from the remaining air sample and catalytically reduced to graphite (Lux, 2018). The graphite targets are then analyzed for the $^{14}C/C$ ratio with an accelerator mass spectrometer at the Curt-Engelhorn-Centre Archaeometry in Mannheim, Germany. Since the measurement process is complex and funding is limited, a thorough selection of appropriate samples is necessary. Several criteria are important to consider.

First, it must be ensured that there were no technical problems with the IRGASON or the flask sampler, i.e., that the
$CO_2$ signal strength of the IRGASON was above 90 %, that the valves were switching, the flow rate into the buffers was



approximately constant and that the horizontal wind direction was not from sectors with known flow distortion effects of the ultrasonic anemometer or the tower structure. Second, the assumptions made in the EC and REA methods, namely stationarity and well-developed turbulence (Sect. 2), must have been fulfilled during the sampling period. For this purpose, the 20 Hz measurements of the IRGASON during the sampling period can be analyzed. In this study, the software EddyPro (Version 250 7.0.9, Licor Inc., Lincoln, NE, USA) was used to process the data. Flask samples were discarded if the integral turbulence characteristics test for $w$ (Foken et al., 2004) was greater than 100 % and the steady state test for the covariance between the vertical wind $w$ and the $CO_2$ concentration (Foken et al., 2004) was greater than 400 %.

Depending on the aim of the application, further selection criteria may be considered in the selection of REA flasks. If the objective is a $^{14}$C-based decomposition of total $CO_2$ surface fluxes, the fossil fuel $CO_2$ differences between updraft and 255 downdraft samples should be greater than the inherent measurement uncertainty of 0.7 ppm that is caused by the current $^{14}CO_2$ measurement precision (see Appendix D). For times in which photosynthesis is expected to be weak, estimates of total $\Delta CO_2$ based on the IRGASON measurements are good indicators.

### 4.5 Zurich installation

The REA system was installed and operated for the first time in Zurich, Switzerland, from July 2022 to April 2023. The 260 IRGASON and REA inlets were mounted on a radio tower of 16.5 m height on top of a 95.3 m tall high-rise residential building (Fig. 3). The data loggers, network devices, and the REA flask sampler were placed in a climate-controlled room at the top floor of the building. The building is surrounded by an industrial sector, railway lines and a busy commuting road to the north, an urban sector (city center) to the south-east and a green, less densely populated area with a cemetery to the south-west. 1 h sampling periods were chosen to match the current resolution of mesoscale inverse models. The deadband was first used 265 with pre-set wind statistics, i.e., adjusted to the mean and standard deviation of the vertical wind speed of the preceeding 30 min period (Table 2). In this case, changes in wind statistics during the sampling period often lead to an unequal volume filled into the updraft and downdraft buffers and on average only every fourth REA sample pair could successfully be transferred to the flask sampler. With a dynamic deadband, implemented and used since the end of October 2022, the success rate increased to about 75 % for the remaining samples. After 12 experimental runs in the beginning of the campaign with varying deadband 270 widths of $\delta = 0.3$, $0.4$ or $0.8$, the deadband width was always set to $\delta = 0.7$ for the remainder of the campaign. In this case, the solenoid valves switched on average $23 \pm 11$ times per minute and air was collected around 50 % of the time. To sample sufficient air for filling a flask (Sect. 4.3), the flow rate into the buffers was $4.67\,\mathrm{l\,min^{-1}}$. With an estimated pump flow velocity of $5 \pm 1.5\,\mathrm{l\,min^{-1}}$ and 33 m long Synflex tubes with inner radius $r_i = 2.65$ mm connecting the inlets with the loop system, the rinse time was set to 8 s (Eq. 9). During this rinse time, the flow rate at the MFCs was only $3.5\,\mathrm{l\,min^{-1}}$. In total, 640 REA runs 275 were started, 300 samples were successfully transferred into flasks and 103 flask pairs were eventually analyzed for greenhouse gases including $^{14}CO_2$.





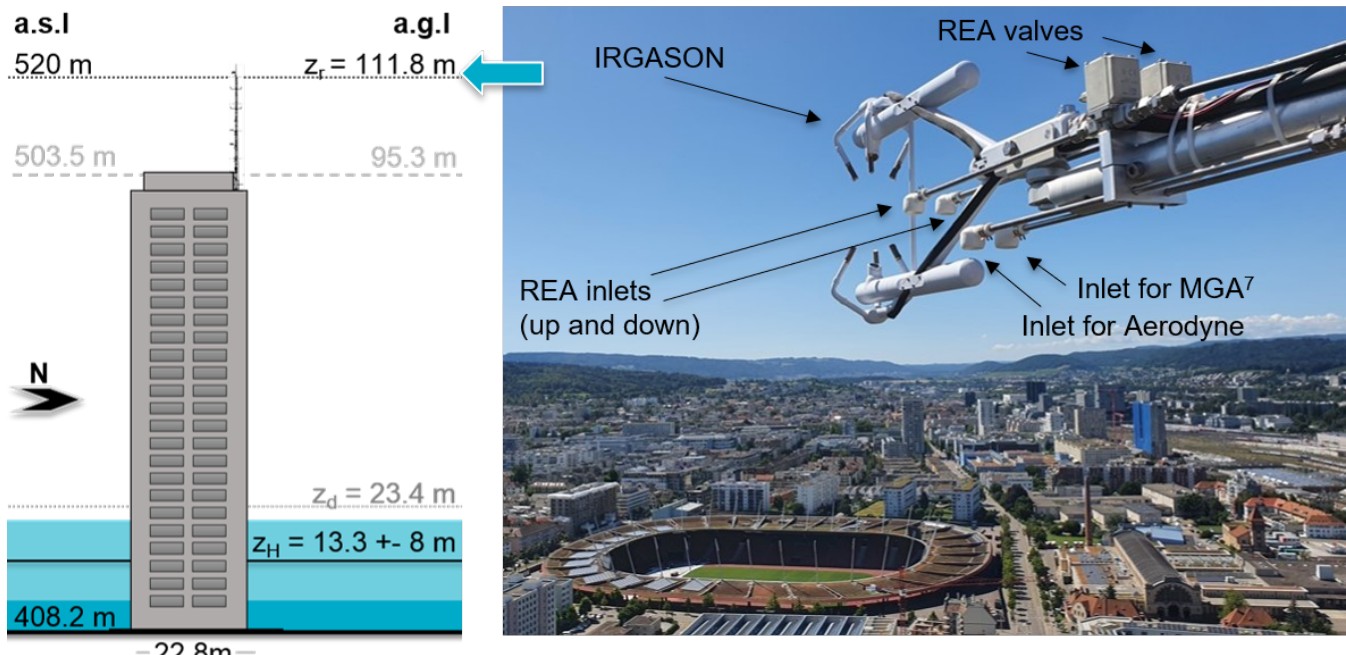

**Figure 3.** Setup of the Zurich campaign. Left: Schematic illustration of the building Hardau II with the mast, where measurements were made. Given are also the corresponding heights in m above sea level (a.s.l.) and above ground level (a.g.l.). $z_H$ denotes the mean building height within 1.5 km radius, $z_d$ is the displacement height according to Kanda et al. (2013). Right: Picture of the IRGASON and the inlets for the relaxed eddy accumulation (REA) system, a multicompound (MGA[7]) gas analyzer and an Aerodyne for COS measurements. The latter were placed in a room on the top floor of the building and connected by 33 m long Synflex tubings.

## 5  Quality control of REA flask samples

The final setup of the REA flask sampling system, including the chosen materials (Table A1) and the flow scheme with the specially designed loop systems (Fig. 1), was the result of many preliminary assessments, simulations, tests and iterative system improvements to meet the high technical requirements of the REA method and the sample analysis. However, due to non-idealities and uncertainties in the sampling procedure, the concentration of the air that is collected in the updraft and downdraft flasks may deviate from the "true" sample concentration that would result from a certain temporal variation of the gas concentration and the vertical wind velocity. To evaluate the performance of the system, i.e., check for biases and quantify the uncertainty of the flask concentrations due to the sampling process, several experiments and simulations were performed.





This section describes:

1. Quality control tests in the laboratory ensuring that the collected air is generally transferred to the glass flasks without contamination

2. Simulations using 20 Hz $CO_2$ in situ measurements in Zurich to estimate biases and uncertainties of $\Delta CO_2$ flask measurements due to non-idealities in the sampling process

Table 3: Mean $\Delta CO_2$ uncertainties

3. Quality control tests performed during the Zurich campaign and comparison of measured flask concentrations with in situ measurements to assess the quality of the samples collected in Zurich

The focus was on the uncertainty of the $CO_2$ concentration differences between updraft and downdraft flasks ($\Delta CO_2$), because
only the concentration differences are needed to calculate fluxes (Eq. 1). Moreover, the flask concentration differences, in contrast to the absolute concentrations, were comparable with high-frequency in situ $CO_2$ measurements of the IRGASON and a multicompound (MGA[7]) gas analyzer (MIRO Analytical AG, Wallisellen, Switzerland), despite an irregular calibration of the gas analyzers. It must be noted that when estimating fossil fuel $CO_2$ differences, uncertainties due to the sampling process are negligible compared to the $^{14}$C measurement uncertainty (compare Sect. 3 and Appendix D).

**5.1 Quality control tests in the laboratory**

Several quality control tests were carried out at the ICOS Flask and Calibration Laboratory in Jena to investigate bias or uncertainty due to contamination and memory effects, i.e., the dependence of the measured concentration on the previous sample, for example due to incomplete evacuation of the buffers. In addition, the rinse time required at the beginning and end of a sampling period was determined by injecting $CO_2$ pulses.
By sampling gas from a cylinder with known concentration under sampling conditions as close to reality as possible (for details see Appendix B), it was shown that neither the buffers, nor the intake lines or the switching of the valves alter the gas concentrations significantly. As shown in Fig. 4, the $CO_2$ concentration of test flasks filled with the sampling system using updraft and downdraft intakes, the loop systems and buffers agree within 1 $\sigma$ with each other and with the cylinder concentration, and mostly meet the WMO compatibility goal for $CO_2$ of 0.1 ppm (WMO recommendation for compatibility
of measurements of greenhouse gases and related tracers (Tans and Zellweger, 2014)).



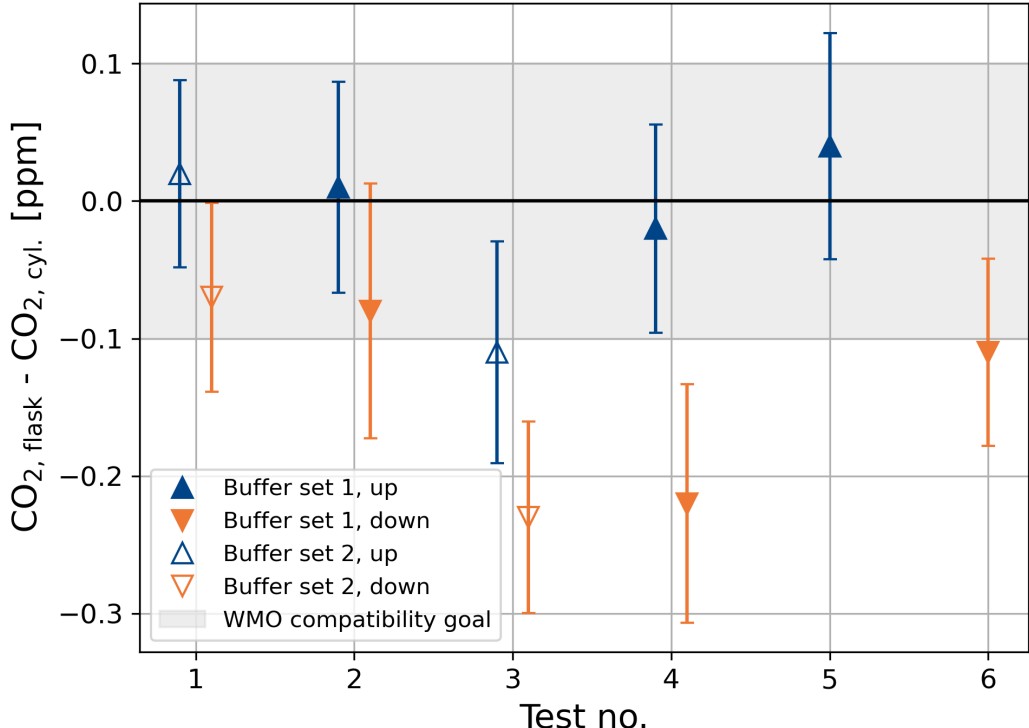

**Figure 4.** Deviation of measured flask sample concentrations from the $CO_2$ concentration of the reference cylinder ($405.71 \pm 0.01$ ppm) that was connected to the fast-response solenoid valves. The gray shaded area highlights the WMO compatibility goal of 0.1 ppm for $CO_2$. Details of the experiments are given in Appendix B.

A memory effect was visible after filling one buffer with pure nitrogen (causing a diluted $CO_2$ concentration of 404.6 ppm measured in the flask compared to 405.8 ppm inserted from the cylinder). However, during the Zurich campaign, the difference between two consecutive buffer fillings was on average approximately $0 \pm 52$ ppm. This means that there is no systematic bias due to the memory effect and that the additional uncertainty is estimated to be about 0.15 ppm, depending on

the $CO_2$ difference to the previous sample. The impact on $\Delta CO_2$ is expected to be even smaller (uncertainty of about 0.02 ppm) since the updraft and downdraft samples are usually affected in a similar way (see Appendix B2).

It was also shown that the time needed to pump an air parcel from the inlet to the buffer estimated from the flow rates and the volumes of the tubes (Eq. 9) agrees well (within less than 1.5 $\sigma$) with the measured travel time of an artificial $CO_2$ pulse. This confirms the calculation of the rinse time according to Eq. (9).

**5.2  $\Delta CO_2$ uncertainty simulations**

The laboratory tests "only" proved the ability to successfully transfer sample air from the REA inlets to the flasks without altering the gas concentration. Several aspects that would affect the measurements in a real REA run, i.e., with a temporally





varying gas concentration, were investigated through simulations using the high-frequency $CO_2$ measurements of the MGA[7]. For this purpose, the 10 Hz time series of the MGA[7] (available since early August 2022, with a few outages, covering 74 REA sampling periods) was upsampled to 20 Hz and synchronized with the IRGASON data by finding the time lag of maximum correlation between the high-frequency $CO_2$ measurements. Mean concentrations of the updraft and the downdraft sample could then be simulated from the high-frequency data using the 20 Hz REA flags (Sect. 4.1). In this section, $\Delta CO_2$ uncertainties and potential biases are assessed solely based on the MGA[7] data. To make the in situ estimates comparable to the air samples that were dried during the transfer to the flasks (compare Sect. 5.3), the measured gas densities were converted to dry air molar fractions (see Appendix C1).

### 5.2.1 $\Delta CO_2$ with delayed collection of air

Time lags between a change of the sign of the vertical wind velocity fluctuations $w' = w - \overline{w}$ and the actual collection of air to the respective reservoirs are a known source of uncertainty in REA flux measurements (e.g., Baker et al., 1992; Pattey et al., 1993). In the given setup, the IRGASON measures the vertical wind speed at a frequency of 20 Hz, so that the maximum delay between the change in wind speed and its detection is 50 ms. As mentioned in Sect. 4.1, the IRGASON output has, at the maximum bandwidth of 20Hz, a time delay of 200 ms. With a bandwidth of 10 Hz, as used by default in Zurich, there is a 400 ms delay. The response time of the solenoid valves is 50 ms or less (open) / 150 ms or less (close). Consequently, it is assumed that the collection of air is delayed by up to 500 ms.

The effect of a delayed collection of air was investigated by calculating the expected $CO_2$ difference between updraft and downdraft sample of the Zurich REA runs based on the 20 Hz REA flags considering different time lags (see Appendix C3). As expected, the larger the delay, the smaller the concentration differences, as air is "sampled" from within the deadband while portions of the larger signals are missed. On average, a 500 ms delay reduces $\Delta CO_2$ by $0.04 \pm 0.06$ ppm. If the delay is 100 ms smaller/larger than expected (i.e., 400 or 600 ms), $\Delta CO_2$ changes by less than 0.02 ppm. More results are given in Table C2.

### 5.2.2 $\Delta CO_2$ with incorrect rinse time

Compared to systems with a single inlet, the direct separation of up- and downdraft sample close to the ultrasonic anemometer of the IRGASON prevents biases due to uncertainties in the travel time of air from the inlet to the buffer and a potential mixing of air in front of the valves. The travel time, however, becomes important at the start and end of sampling where the rinse time determines the opening and closing of the magnet valves at the buffers (Sect. 4.2). Given the uncertainties in the pump speeds and the lengths of the tubing, the calculated rinse time (Eq. 9) in the Zurich setup had an uncertainty of about 2 s. This can lead to sampling of unwanted air and loss of wanted air. The impact on the concentration difference between the updraft and downdraft flasks was estimated from MGA[7] data by discarding the first two seconds of measurements in sampling mode and adding another two seconds after the sampling end and vice versa (see Appendix C4). There is no systematic bias and the standard deviation in the change of $\Delta CO_2$ is 0.01 ppm.



### 5.2.3 $\Delta CO_2$ with variable sampling rate


Another source of uncertainty is the actual flow rate at which the air is sampled. This flow rate should be constant to ensure a homogeneous weighting of the collected air over time. For this purpose, mass flow controllers were placed right before the buffers. However, the MFCs are slightly affected by humidity and have a response time of 1 to 2 s. This means that after a change in the pressure difference across the element it will take up to 2 s for the flow to become constant again. In the REA

system, the valves can switch with a frequency of up to 10 Hz and with each switching the pressure behind the MFC changes between atmospheric pressure (standby mode) and the pressure in the buffer (sampling mode). Consequently, the deviation from the desired flow is especially large at the beginning and end of a sampling period, when the pressure in the buffer is 1 mbar or up to 1600 mbar, respectively. In addition, unsynchronized switching of the top valves (20 Hz scan rate) compared to the valves in the loop systems (10 Hz scan rate) can lead to underpressure in the intake line. The greater the underpressure,

the higher the flow rate at the inlet when the top valve opens the next time. At the beginning of the Zurich campaign, there were additional biases due to a difference in flow rate during the rinse time compared to the rest of the sampling period and due to a loss of sample when one of the two buffers reached the maximum pressure of 1.6 bar.

To estimate the effect on the concentration differences between updraft and downdraft flasks, $\Delta CO_2$ was simulated 103 times for each REA run during the Zurich campaign, each time with a different weighting of the MGA[7] or IRGASON mea-

surements. The weighting was thereby based on actual flow rate measurements of the MFCs (see Appendix C5). While there is no systematic bias, the standard deviation of the $\Delta CO_2$ difference between an inhomogeneous and a homogeneous weighting is on average $0.03 \pm 0.05$ ppm. The deviations are largest at sampling periods with high $CO_2$ variability, where the absolute $\Delta CO_2$ is also usually larger than average (see Fig. C1). Consequently, the relative uncertainties are less affected. To minimize this uncertainty, the flow rate during the rinse time was adjusted to the flow rate during the sampling period, the maximum

buffer pressure, i.e., the pressure at which sampling is stopped, was reduced to 1.55 bar and an additional pressure regulator was installed in each loop system.

### Mean $\Delta CO_2$ uncertainties

Table 3 summarizes the estimated $\Delta CO_2$ biases and uncertainties due to the different aspects mentioned above. It is important to note that the given means and standard deviations are solely based on data from the 103 REA runs during the Zurich

campaign and a small number of quality control tests. The results may be different for other time periods and other sites. However, the estimates show that except for a delayed collection of air, there is no bias in the concentration differences between updraft and downdraft sample. The standard deviations of the data set, on the other side, are of the order of the $\Delta CO_2$ measurement uncertainty at the GC. This means that for individual samples, e.g., those collected during high variability in the $CO_2$ concentration of the ambient air, the flow variability and other non-idealities in the sampling process can be significant

sources of $\Delta CO_2$ uncertainty. Regular quality control tests and comparisons of flask samples with in situ measurement data are therefore important for independent validation of the performance of the system during a measurement campaign (see Sect.





5.3). Due to dependence on the ambient sampling conditions, the different $\Delta CO_2$ uncertainty contributions should in this case be considered for each sampling period individually.

To estimate ffCO$_2$ differences on the contrary, the measurement uncertainty of $^{14}CO_2$ is by far the dominant source of

uncertainty (see Appendix D). Moreover, when estimating ffCO$_2$ fluxes, $\beta$ calculated from CO$_2$ for each REA run individually, accounts for most of the above-mentioned effects (Pattey et al., 1993). In this case, the uncertainties due to the sampling process are considered negligible.

**Table 3.** Overview of the estimated biases and uncertainties in the $CO_2$ concentration difference between updraft and downdraft flasks $\Delta CO_2$ due to non-idealities in the sampling procedure. Given are the means (bias) and standard deviations (uncertainty) of the measured or simulated changes in $\Delta CO_2$ with respect to the expected values. For completeness, the measurement uncertainty of the gas chromatographic analysis of the flasks is also given. $N$ represent the number of measurements or sampling periods of the underlying data set.

| Source of $\Delta CO_2$ bias / uncertainty | Data sets | $N$ | $\Delta CO_2$ bias [ppm] | $\Delta CO_2$ uncertainty [ppm] |
|---|---|---|---|---|
| Memory and surface effects | Lab measurements (flasks) | 2 | - | 0.02 |
| | IRGASON-based CO$_2$ estimates of subsequent REA samples | 738 | | |
| 500 ms delay in collection of air | 20 Hz CO$_2$ from MGA[7] | 74 periods | -0.04 | 0.06 |
| | 20 Hz REA flags from IRGASON | | | |
| 2 s uncertainty in rinse time | 20 Hz CO$_2$ from MGA[7] | 74 periods | - | 0.01 |
| | 20 Hz REA flags from IRGASON | | | |
| Variable flow rate | 20 Hz CO$_2$ from IRGASON / MGA[7] | 103 periods | - | 0.03 |
| | 20 Hz REA flags from IRGASON | 103 periods | | |
| | 0.2 Hz flow rate measurements | 103 periods | | |
| GC measurement uncertainty | Lab measurements (flasks) | 103 flask pairs | - | 0.04 |



### 5.3 Quality control tests during the campaign

To ensure that the REA flask sampling system was working as intended, quality control tests were performed regularly through-
out the Zurich campaign. In addition, the measured concentration differences of the REA flask pairs were compared with the
in situ measurements of the IRGASON and the MGA[7].

#### 5.3.1 All-valves-open tests

To check for biases between updraft and downdraft sampling, as well as for leaks or other sources of contamination, "all-
valves-open" tests were performed about once a month. This involved continuously filling two buffers with ambient air by
turning on the pumps and opening both solenoid valves at the inlets as well as the valves in the loop systems until a buffer
pressure of about $1.4$ bar was reached and the samples could be transferred to the flask sampler. Biases between updraft and
downdraft lines would result in concentration differences between the two corresponding flasks. To detect systematic errors
that might affect both lines equally, a third flask was sampled simultaneously with a $1/t$ flow rate (Levin et al., 2020) through a
separate tubing directly into the flask sampler, bypassing the REA loops and the buffers. If the system is working as expected,
the concentrations in the three flasks should agree within the WMO compatibility goal. Another quality control would be to
compare the absolute concentrations of the flasks with the average concentration of in situ measurements over the sampling
period. In Zurich, however, no accurate measurements of ambient air near the inlets were available (neither the IRGASON nor
the MGA[7] were meant to be calibrated regularly according to WMO standards).

During the Zurich campaign, nine all-valves-open tests were performed. Figure 5 shows the $CO_2$ concentration of each
flask compared to the respective mean of the "up" and "down" samples. In tests 2, 3, 7, 8, and 9, an additional flask was
sampled directly into the flask sampler. However, it was later found that, at least at the beginning of the sampling periods, the
flow rate was lower than intended. This means that the air in the direct flasks could not be exchanged sufficiently, so that the
concentration does not represent the actual mean value, but is also influenced from the purging period preceding the sampling.
This particularly affects the flasks that were collected during a period with a large variability in concentration.

It can be seen that for tests 1, 2, 3, 4, 8, and 9, up and down flasks agree within their measurement uncertainties, indicating
that there is no significant bias between the two lines. The difference between the pairs of simultaneously collected flasks is
on average $-0.01 \pm 0.03$ ppm. The smaller $CO_2$ concentrations of the direct flasks of tests 8 and 9 can be explained by a
large $CO_2$ variability with an overall increase over time. The fact that in tests 2 and 3, up and down flasks also agree with the
direct sample within the WMO compatibility goal indicates that there was no bias from the loop system or from the buffers
that would have affected up and down in the same way.

In tests 5, 6, and 7, on the contrary, there are large concentration differences between up, down and direct flask (note the
different scale on the y-axis). This indicates a leak in the system, which also explains the large deviations of the measured
$CO_2$ concentrations from the expected $CO_2$ concentrations based on in situ measurements observed at this time (compare next
section). Unfortunately, due to the long time lag between sampling and the availability of concentration measurements, the
leak could only be detected and fixed after several months.



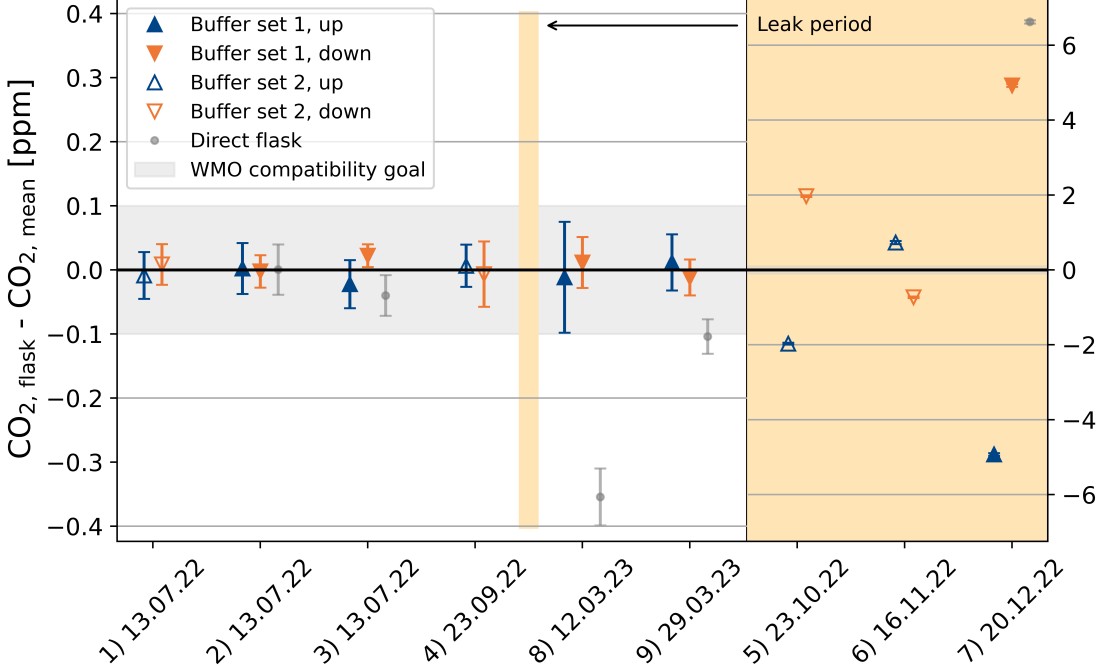

**Figure 5.** Comparison of flask concentrations from all-valves-open tests. Shown are the differences in $CO_2$ between flasks sampled through the updraft and downdraft intakes of the REA system and a flask sampled directly into the flask sampler compared to the mean of the two samples from the REA intake system. Large deviations in tests 5, 6 and 7 (beige shaded parts with different scale on the y-axis) were caused by two leakages in the loop systems. Outside this period, the results mostly agree within the WMO compatibility goals.

In summary, the results show that in general there is no significant bias between updraft and downdraft sampling and that all-valves-open tests help detect leaks.

### 5.3.2 Flask - in situ comparison

In addition to the all-valves-open tests, the measured concentration differences of the REA flask pairs were compared to in situ

measurements from the IRGASON and the co-located MGA[7] by averaging the high-frequency data from the periods during which the respective valves were open and air was sampled, as denoted by the 20 Hz REA flags (see Appendix C2). For the final $\Delta CO_2$ estimates an uncertainty of $\sqrt{2} \cdot 0.15$ ppm for the IRGASON estimates and $\sqrt{2} \cdot 0.1$ ppm for the MGA[7] estimates was assumed, based on the precision of the instruments stated by the manufacturers. For the flask data uncertainty, the GC measurement uncertainties as well as the individually simulated sampling uncertainties from a memory effect, an uncertainty

in the time lag between the vertical wind signal and the conditional collection of air, an uncertainty in the rinse time, and an inhomogeneous weighting of sample air due to variability in the sampling rate were taken into account (compare Table 3).

The results of 102 REA runs during the Zurich campaign are shown in Fig. 6, which plots the differences between $\Delta CO_2$ from flask measurements and in situ estimates from the IRGASON and the MGA[7] over the sampling time (for one REA





run, neither IRGASON nor MGA[7] data are available). The samples from November 2022 to February 2023, which were
contaminated due to a leak (Sect. 5.3), were discarded.

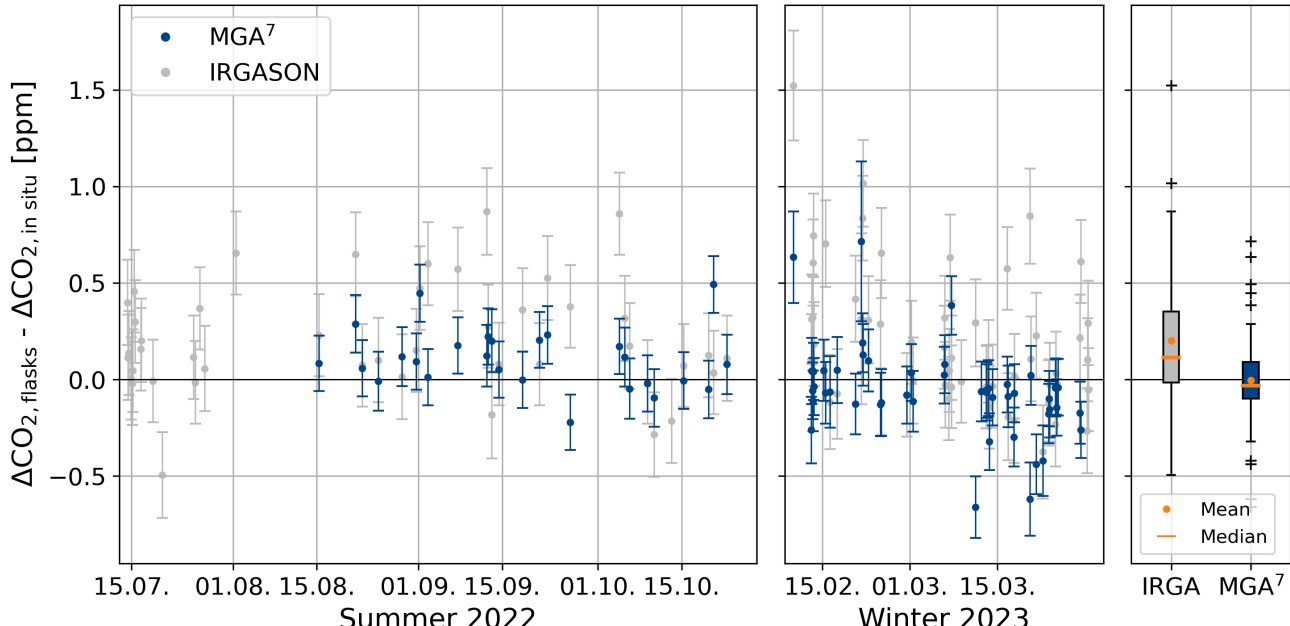

**Figure 6.** Difference between $\Delta CO_2$ (up-down) measured in the flasks and estimated from high-frequency measurements of the IRGASON
and the MGA[7]. Error bars represent the flask measurement uncertainty (analysis plus estimated uncertainty due to the sampling process) and
an estimated uncertainty of 0.15 ppm for the IRGASON and 0.1 ppm for the MGA[7]. MGA[7] data are only available since mid of August.
The leak period between end of October and beginning of February, as well as measurements of the IRGASON with $CO_2$ signalstrength
< 90 % and correlation with the MGA[7] < 0.5 were excluded. The box plot on the right shows the mean, median and interquartile range of all
summer and winter samples.

The fact that the MGA[7] estimates agree very well with the flask measurements (mean difference in $\Delta CO_2$ < 0.01 ppm
with a standard deviation of 0.23 ppm) provides evidence that there were no major leaks or significant biases due to the
sampling process. A potential bias due to a delayed collection of air (Sect. 5.2.1) is therefore considered negligible. Four
$\Delta CO_2$ measurements deviated from the expected value by more than 0.5 ppm. The exact reasons are not known, but for
example, during the REA run on 21 February 2023 (7:00–8:00 LT), the $CO_2$ concentration increased by more than 100 ppm
and the $CO_2$ difference between updraft and downdraft sample was the highest observed. During the REA run on 11 March
2023 (13:30–14:30 LT), the wind measurements were very spiky, likely due to a rain event earlier in the day, and according to
the REA flags, the valves did not switch as one would expect from the wind data. These sampling periods must therefore be
examined more closely before further analysis.




The IRGASON-based $\Delta CO_2$ estimates deviate significantly more from the flask measurements with an average of $0.20 \pm 0.33\,\mathrm{ppm}$. This, however, is most likely linked to the fact that the IRGASON $CO_2$ dry molar fractions were derived from a $CO_2$ density output that does not properly account for high-frequency fluctuations in air temperature in the sensing path, because the ambient temperature measured by the EC100 slow-response temperature probe is used in the conversion of absorption measurements to $CO_2$ density (see Appendix C1). As described by Helbig et al. (2016), this causes a systematic bias compared to closed-path gas analyzers due to the high-frequency temperature attenuation. Indeed, the difference between $\Delta CO_{2,\mathrm{flasks}}$ and $\Delta CO_{2,\mathrm{IRGA}}$ correlates with the difference between ultrasonic temperature during updraft and downdraft conditions. Helbig et al. (2016) showed that this bias could be reduced significantly by using the ultrasonic anemometer's fast-response temperature. Unfortunately, this additional $CO_2$ density output, available in EC100 OS version 7.01 or later, was not recorded during the Zurich campaign. Since the differences between IRGASON and flask measurements could be explained by the insufficient correction of spectroscopic effects during high sensible heat fluxes, the good agreement between flask data and MGA[7] measurements indicates an overall successful implementation of the REA method.

## 6 $\Delta CO_2$ partitioning

Figure 7 shows the differences in $CO_2$ and ff$CO_2$ between the updraft and downdraft flasks, measured with the gas chromatograph at the ICOS FCL in Jena and derived from the $^{14}CO_2$ analyses at the ICOS CRL in Heidelberg (Eq. 8). Of the 103 selected REA flask pairs, three samples were lost during graphitization, eight sample pairs from the end of the campaign were not analyzed for $^{14}CO_2$ due to their small total $CO_2$ difference and four sampling periods were subsequently considered unsuitable for REA measurements due to lack of stationarity or corrupt wind measurements, leaving a total of 88 $\Delta$ff$CO_2$ estimates. The error bars represent the measurement uncertainties ($\Delta CO_2$ uncertainties of about $0.05\,\mathrm{ppm}$ in x-direction are omitted for clarity). In the analysis of ff$CO_2$, uncertainties due to the sampling process were considered negligible (Sect. 5). The colors indicate the month in which the sample was collected.

It can be seen that the largest concentration differences between updraft and downdraft flasks with $\Delta CO_2$ of up to 13.6 ppm were collected in February and March, when anthropogenic emissions are usually higher than average due to residential heating. Most of these samples are indeed close to the 1:1 line, which represents the case where the measured $CO_2$ fluxes are completely due to fossil fuel emissions. Accordingly, the observed deviations from this line to the right or the left indicate respiratory and other non-fossil sources or photosynthetic signals, respectively. In agreement with other studies (e.g., Wu et al., 2022; Crawford and Christen, 2015), this shows that there are significant non-fossil $CO_2$ signals even in winter. It should be noted that the applied sample selection criteria do not necessarily exclude sampling periods in which a change in the storage below the sampling height contributes to the measured fluxes. This is the case, for example, when the depth of the atmospheric boundary layer increases rapidly due to convective, turbulent vertical motions generated by radiative heating of the surface in the morning (Stull, 1988). Based on an observed drop in $CO_2$ concentration in the morning and a preliminary investigation of the turbulence structure of the atmosphere, some of the Zurich REA flasks with large $CO_2$ concentration differences between updraft and downdraft sample were probably collected during this time. In this case, the composition of




the samples does not necessarily represent the actual surface fluxes during the REA sampling periods, but rather the integrated nocturnal fluxes. Further interpretation of the results and a subsequent investigation of the corresponding $CO_2$ fluxes therefore
requires a thorough analysis of the sampling periods.

What also becomes clear in Fig. 7 is that most signals are of the order of 1 ppm or less, which is small compared to the $\Delta ffCO_2$ uncertainty. The latter was on average 1.2 ppm and primarily due to the current $^{14}CO_2$ measurement precision (Appendix D). ffCO$_2$ flux estimates derived from samples with $\Delta ffCO_2 < 1.2$ ppm will therefore have uncertainties of more than 100 % (compare Eq. 2).

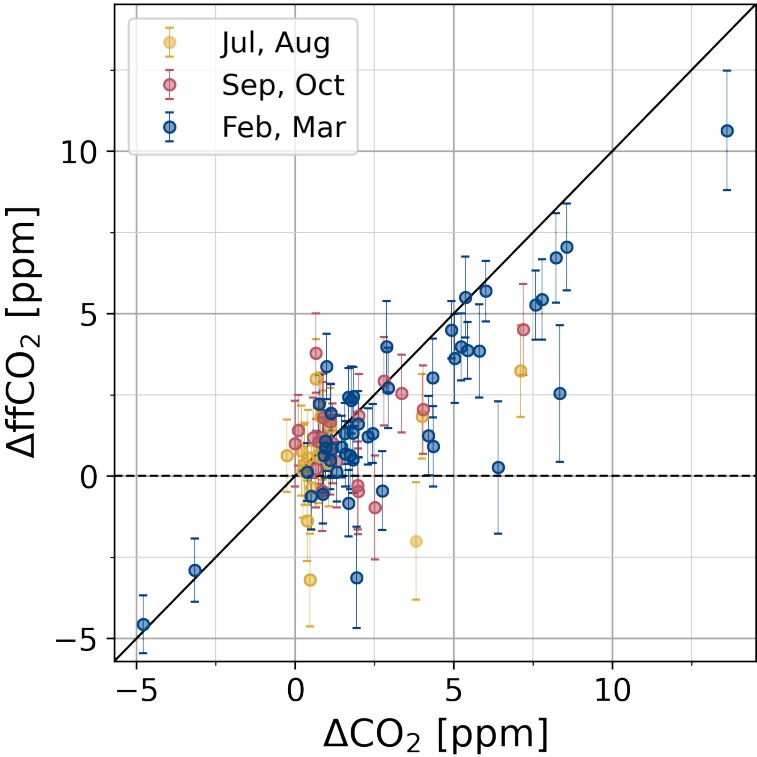

**Figure 7.** $\Delta ffCO_2$ and $\Delta CO_2$ of the 88 selected REA flask pairs from Zurich.

**7 Conclusions**

A relaxed eddy accumulation (REA) flask sampling system for $^{14}$C-based estimation of ffCO$_2$ fluxes for urban eddy covariance (EC) flux measurements was developed and tested in the ICOS Flask and Calibration Laboratory in Jena, Germany, and on a tall-tower near the city center of Zurich, Switzerland.

Two fast-response valves are activated by the vertical wind signal from an ultrasonic anemometer of a co-located IRGASON.
The conditionally collected air accumulates in two separate cylinders (one for the updraft, one for the downdraft sample) and



is transferred to 3 l glass flasks after the sampling period using an automated 24-port flask sampler. The samples can thus be analyzed in the laboratory for a variety of gases, including $^{14}CO_2$. Laboratory tests and quality control tests during the first measurement campaign in Zurich showed that the sampling procedure does not cause a significant bias in the $CO_2$ differences between updraft and downdraft samples. Uncertainties due to the sampling process are negligible when estimating fossil fuel

$CO_2$ differences between the updraft and downdraft flasks and consequently ffCO$_2$ fluxes, as these are dominated by the $^{14}CO_2$ measurement precision. The novel REA flask sampling system itself fulfills the high technical requirements of relaxed eddy accumulation, providing high quality data for scientific studies of multiple non-reactive species.

Due to the prerequisites of the EC and REA method, e.g., stationarity and well-developed turbulence, and the costs and efforts associated with the flask analysis, only a limited number of individual sampling periods can be analyzed. In general,

operating the system, i.e., scheduling sampling events, analyzing and selecting suitable sampling periods, sending the flasks to the laboratory, etc., requires frequent remote and on-site work.

Given the good agreement between the total $CO_2$ concentration differences measured in the REA flask sample pairs and observed from in situ measurements of a MGA[7], the results of the first measurement campaign in Zurich serve as a proof of concept for a $^{14}C$-based separation of fossil and non-fossil $CO_2$ signals. As expected, the $CO_2$ differences between updraft and

downdraft sample were largest during the heating season in February and March. In this case, fossil fuel emissions were the major contributor. However, even in winter, small photosynthetic and significant non-fossil (respiration and biofuels) signals were observed, highlighting the role of the biosphere in an urban environment.

The main challenge so far was a generally small signal-to-noise ratio of measured $^{14}CO_2$ differences. Resulting $\Delta$ffCO$_2$ uncertainties on the order of 100 % limit the interpretation of individual results. Two improvements are proposed to potentially

increase the concentration differences in future campaigns: First, the two pumps in the loop systems are recommended to be replaced by larger pumps with a higher pump speed, allowing a reduction in the proportion of time that air is collected and thus a larger deadband width $\delta$. Second, the so-called hyperbolic relaxed eddy accumulation could be added in the logger program. With this setting, proposed by Bowling et al. (1999), air would only be collected if the fluctuations in vertical wind speed $w'$ and concentration $c'$ are above a certain threshold. For the latter, the 20 Hz $CO_2$ measurements of the IRGASON

could be used. Excluding eddies with little flux contribution increases the concentration differences more effectively than a linear deadband with larger deadband width and is recommended for scalars where the detection limit is of concern (Vogl et al., 2021). However, the hyperbolic deadband comes with additional challenges, as the threshold for not only vertical wind velocity fluctuations, but also for $CO_2$ mixing ratio fluctuations would need to be dynamically adjusted.

The next step in the process will be to derive the actual ffCO$_2$ fluxes using EC-based total $CO_2$ fluxes and assuming scalar

similarity to estimate the $\beta$ coefficient for each sampling period individually (Eq. 2). This will allow a fully independent time-resolved evaluation of the Zurich emission inventories, taking into account the changing turbulent flux footprints during each of the sampling intervals. Furthermore, the analysis of CO and other species in the updraft and downdraft samples along $^{14}CO_2$ will provide guidance on the use of co-emitted species for partitioning total $CO_2$ fluxes into fossil and non-fossil components.



*Code and data availability.* The logger program and the data supporting this publication are provided at https://doi.org/10.5281/zenodo.
525    13926680.

## Appendix A:  Setup of the REA flask sampling system

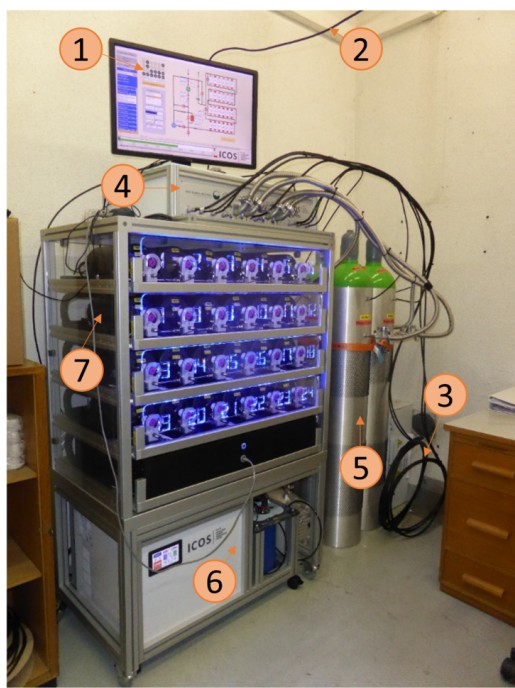

**Figure A1.** Photo of the REA flask sampling system as installed in the control room below the tower at Zurich-Hardau. 1: Screen with flask sampler software; 2: Connection to the CR6 data logger; 3: Intake lines; 4: Loop system; 5: Buffers; 6: Air dryer; 7: Flasks





**Table A1.** Components of the REA flask sampling system (compare Fig. 1 and Fig. 2).

| Abbreviation | Instrument | Company | Model (Version) |
|---|---|---|---|
| IRGASON | Infrared gas analyzer and 3D ultrasonic anemometer | Campbell Sci. [a] | IRGASON (SS2-BB-IC) |
| DC controller | 8 channel solid state DC controller | Campbell Sci. [a] | SDM-CD8S |
| IRGASON electronics | Gas analyzer electronics with enclosure | Campbell Sci. [a] | EC100 |
| Data logger | Measurement and control datalogger | Campbell Sci. [a] | CR6 |
| Pump (Fig. 1) | Pump to transfer samples to flasks | KNF [b] | N816AV.12DCB |
| | Pump to evacuate buffers | Edwards [c] | nXDS6i / Edwards nXDS15i |
| Pump (Fig. 2) | Pump to transfer air into buffers | KNF [b] | N816AV.12DCB |
| MFC | Mass flow controller 0.2-10 $l_n$ min$^{-1}$ | Bronkhorst [d] | F-201CV-10K-AAD-22-V |
| Pressure control valve | Pressure control valve | SMC [d] | AP100-N02B-X201 |
| Valve in the loop system | Three port magnet valve | SMC [d] | VX3244HZ-02N-5DS1-B |
| Valve at the buffer | Vacuum magnet valve | SMC [d] | XSA3-32S-5D2 |
| Solenoid valve at the inlet | Solenoid valve | SMC [d] | VX214NFB |
| $p_1$ - $p_4$ | Pressure sensors | SMC [d] | PSE543A-N01 |
| Buffer 1 - 4 | Cylinders | Matar [e] | 50 l aluminum cylinder |

[a]: Campbell Scientific Inc., Logan, UT, USA; [b]: KNF Neuberger GmbH, Freiburg, Germany; [c]: Edwards Vacuum, Burgess Hill, UK; [d]: SMC Corporation, Tokio, Japan; [e]: Matar, Mazzano, Italy.

## Appendix B: Laboratory tests

### B1 Contamination test 1

To test for a potential contamination of the samples during the sampling process, e.g., due to a leak in the line or at the valves
or due to surface effects with the surfaces of tubes and buffers, the REA system was set up in the laboratory of the ICOS FCL in Jena. For practical reasons, the IRGASON as well as the fast-response valves were placed in the same room as the buffers and the flask sampler, but were connected through two > 50 m long Synflex tubes. A reference gas cylinder with $407.71 \pm 0.01$ ppm $CO_2$ was connected to a water trap operated in reversed order. In this way, the humidity of the gas was comparable to that of typical ambient air (dewpoint of approximately 12 °C), avoiding increased surface effects along the tube and in the buffers.
The humidified gas was then directed through a three-way valve to the solenoid valves and sampled into the respective buffers according to a wind signal from the IRGASON, artificially generated through a fan. From there, the gas was transferred into the flask sampler as during a regular REA run. The test was repeated five times, with slightly different setups as listed in Table B1. The results are given in Fig. 4.



**Table B1.** List of differences between the six setups for contamination tests in the laboratory.

| Test no | Setup |
|---------|-------|
| 1 | Open split between humidifier and three-way valve |
|   | Previous sample: cylinder air |
| 2 | Open split between humidifier and three-way valve |
|   | Previous sample: cylinder air |
| 3 | Humidifier directly connected to three-way valve |
|   | Previous sample: cylinder air |
| 4 | Humidifier directly connected to three-way valve |
|   | Previous sample: ambient (room) air |
| 5 | Humidifier directly connected to the solenoid valve on the updraft side, |
|   | while ambient (room) air was sampled on the downdraft side |
|   | Previous sample: cylinder air |
| 6 | Humidifier directly connected to the solenoid valve on the downdraft |
|   | side, while ambient (room) air was sampled on the updraft side |
|   | Previous sample: cylinder air |

## B2 Contamination test 2: Memory effect test

Although the first set of contamination tests (Sect. B1) had already shown that the concentration of the sample is not significantly changed by the sampling process, a second experiment was performed to quantify the influence of the previous sampling on the flask concentration. For this purpose, a cylinder with known gas concentration and pure nitrogen were alternately connected to the loop systems via an approximately 80 m long Synflex tube with 8 mm outer diameter. As in first contamination tests (Sect. B1), the dry gas from the tanks was humidified through a reversed water trap to a dew point of approximately 12

°C, comparable to normal sampling conditions of ambient (humid) air. All valves were opened and one buffer (always buffer 4) was filled six times to approximately 1.2 bar. The cylinder gas was then transferred into the flask sampler and the buffer was evacuated again (residual buffer pressure of about 0.6 mbar). The $CO_2$ concentrations measured in the respective flasks are given in Table B2.





**Table B2.** $CO_2$ concentrations of the flasks compared to the reference gas cylinder that was connected to the inlets, before and after sampling pure nitrogen. The results are given in chronological order of the fillings.

| Filling | $CO_{2,\,cyl}$ [ppm] | $CO_{2,\,flask}$ [ppm] | $|CO_{2,\,flask}\,/\,CO_{2,\,cyl}$ - 1| [%] |
|---|---|---|---|
| 1. Cylinder | $405.75 \pm 0.05$ | $405.83 \pm 0.06$ | $0.020 \pm 0.018$ |
| 2. Nitrogen | 0 | - | - |
| 3. Cylinder | $405.75 \pm 0.05$ | $404.41 \pm 0.05$ | $0.330 \pm 0.017$ |
| 4. Cylinder | $405.75 \pm 0.05$ | $405.67 \pm 0.08$ | $0.020 \pm 0.023$ |
| 5. Nitrogen | 0 | - | - |
| 6. Cylinder | $405.75 \pm 0.05$ | $404.78 \pm 0.05$ | $0.239 \pm 0.017$ |

While the $CO_2$ concentrations of flasks 1 and 4 agreed with the cylinder gas within 1.5 $\sigma$, flasks 3 and 6 collected after
flushing the system with nitrogen, i.e., after an intentionally chosen large $CO_2$ change of 405.7 ppm, deviated by more than 1
ppm. Assuming that the bias in the flask concentration $c_{meas} - c_{ref}$ (measured minus reference concentration) depends linearly
on the concentration difference between the current and the previous sample $c_{ref} - c_{ref,\,-1}$, it can therefore be expected that
$CO_{2meas} - CO_{2ref} = (0.0028 \pm 0.0005) \cdot CO_{2ref} - CO_{2ref,\,-1}$. As uncertainty, half of the difference between the two measurements
was taken. Based on the IRGASON-derived $CO_2$ estimates of updraft and downdraft samples of all REA sampling periods
during the Zurich campaign (702 in total), the mean concentration difference between subsequent buffer fillings was estimated
to be $0 \pm 52$ ppm (assuming that buffer sets 1 and 2 are always used alternately). Consequently, there is no systematic bias in
absolute flask concentrations while the estimated uncertainty is 0.17 ppm.

Since the updraft and downdraft samples are usually affected in a similar way, the effect on concentration differences between
them is expected to be even smaller. Multiplying the mean difference between subsequent concentration differences between
updraft and downdraft sampling (estimated again from the IRGASON data) of $0 \pm 7$ ppm with the $0.28 \pm 0.05$ % derived from
the laboratory experiments (Table B2) results in an estimated uncertainty of about 0.02 ppm. Again, there is no systematic
bias.

**B3   Rinse time measurements**

In the field, the rinse time, i.e., the time between the start (end) of the sampling period and opening (closing) of the valves at
the buffers, is calculated from the estimated pump speed and the dimensions of the intake lines according to Eq. (9). To test
the validity of this approach and quantify the associated uncertainty, a cylinder with pure $CO_2$ was connected at the normally
closed position of a three-way valve installed in front of the fast-response valves. The loop system was in sampling mode
(pump on, three-way valves in the loop systems connecting the pump with the outflow). Then, the three-way valve was opened
to the $CO_2$ cylinder for 200 ms, injecting a short $CO_2$ pulse into the line. This pulse was detected at the outflow of the REA
sampler by a $CO_2$ sensor (SprintIR-WF-20, Gas Sensing Solutions Ltd, Cumbernauld, UK). The travel time of the $CO_2$ pulse
was thus estimated as the time between the opening of the three-way valve and the start of the detected spike. The experiment



was repeated several times and for different tubes (see Table B3). The measurements $t_{r,\text{meas.}}$ were then compared to the values $t_{r,\text{est.}}$ estimated from the length and inner radius of the tube and the flow rate measured at the outflow of the REA sampler. As shown in Table B3, both values agreed within less than $1\,\sigma$.

**Table B3.** Length $l_t$ and inner diameter $r_t$ of the tubes connecting the REA inlets with the loop systems. Together with the the pump flow velocity $q_{\text{pump}}$ (flow rate measured at the outflow), the travel time from the inlet to the outflow $t_{r,\text{est.}}$ was estimated according to Eq. (9) and compared with $t_{r,\text{meas.}}$, which was measured with a $CO_2$ pulse.

| $l_t$ [m] | $r_t$ [mm] | $q_{\text{pump}}$ [l/min] | $t_{r,\text{est.}}$ [s] | $t_{r,\text{meas.}}$ [s] | Deviation $t_{r,\text{est.}}$ and $t_{r,\text{meas.}}$ |
|---|---|---|---|---|---|
| $10.45 \pm 0.10$ | $2.2 \pm 0.1$ | $5.9 \pm 0.5$ | $1.73 \pm 0.21$ | $1.9 \pm 0.2$ | $0.6\,\sigma$ |
| $58.0 \pm 0.1$ | $2.2 \pm 0.1$ | $4.0 \pm 0.5$ | $13.0 \pm 2.0$ | $11.5 \pm 0.2$ | $0.7\,\sigma$ |

The experiment was also done when the REA valves opened or closed once per second and were therefore only open half the time. As expected, the time until the signal was detected was approximately twice as long as when the valves were permanently open. This suggests that the concept of the rinse time also works well at the start of a REA run when the valves are already switching according to the wind signal.

## Appendix C: $\Delta CO_2$ simulations

For uncertainty estimation and quality control, the expected $CO_2$ differences between the updraft and downdraft samples collected in Zurich were calculated from the high-frequency in situ measurements of the IRGASON and the MGA[7]. This required the synchronization between IRGASON and MGA[7] data and the start and end times of the REA runs, despiking of the 20 Hz time series according to Mauder et al. (2013) and conversion of measured gas densities to dry molar fractions. The high-frequency measurements were then averaged over the time periods when the updraft / downdraft valve was open and air
was collected using the 20 Hz REA flags (Sect. 4.1).

### C1   Conversion to dry molar fractions

Flask "concentrations" measured at the gas chromatograph are given in moles of gas per mole of dry air, i.e., molar mixing ratio. To compare them with in situ measurements, the 20 Hz humid air mole fractions $\chi_{\text{gas}}$ recorded by the MGA[7] and the mole densities $d_{\text{gas}}$ from the IRGASON were converted to dry air molar mixing ratios as follows (compare Mauder et al. (2021),
LI-COR (2021)):

   – MGA[7]:

$$r_{\text{gas}} = \frac{\chi_{\text{gas}}}{1 - \chi_{H_2O}} \tag{C1}$$





– IRGASON:

$$r_{\text{gas}} = d_{\text{gas}} \frac{v_a}{1 - \chi_{\text{H}_2\text{O}}} \tag{C2}$$

$v_a = \frac{R \cdot T_a}{p}$ is the ambient air molar volume. Following Helbig et al. (2016), the ambient air temperature $T_a$ is derived from the fast-response ultrasonic temperature corrected for humidity effects (Schotanus et al., 1983):

$$T_a = \frac{T_s}{1 + 0.51 \cdot q} \tag{C3}$$

$$q = \frac{\rho_{\text{H}_2\text{O}}}{\rho_a} = \frac{\rho_{\text{H}_2\text{O}}}{\rho_{\text{H}_2\text{O}} + \rho_d} \tag{C4}$$

$$\rho_d = \frac{p_a - e}{T \cdot R_d} \tag{C5}$$

$$e = \rho_{\text{H}_2\text{O}} \cdot R_v \cdot T \tag{C6}$$

For calculating the specific humidity $q$, the slow-response air temperature $T$ measured by a co-located EC100 thermistor at a resolution of 1 s is used.

The various variables are described in Table C1. Note that for flux calculations in EddyPro, the raw, high-frequency measurements were not converted from molar densities to mixing ratios, but the WPL correction according to Webb, Pearman and Leuning Webb et al. (1980) was applied.

**Table C1.**

| Variable | Unit | Description |
|---|---|---|
| $r_{\text{gas}}$ | mol mol$^{-1}$ | Molar mixing ratio (moles of gas per mole of dry air) |
| $\chi_{\text{gas}}$ | mol mol$^{-1}$ | Mole fraction (moles of gas per mole of wet air) |
| $d_{\text{gas}}$ | mol m$^{-3}$ | Mole density (moles of gas per unit of volume) |
| $v_a$ | m$^3$ mol$^{-1}$ | Ambient air molar volume |
| $T_a$ | K | Ambient air temperature |
| $T_s$ | K | Ultrasonic temperature |
| $q$ | kg kg$^{-1}$ | Specific humidity |
| $\rho_{\text{H}_2\text{O}}$ | g m$^{-3}$ | Density of water vapor |
| $\rho_d$ | g m$^{-3}$ | Density of dry air |
| $\rho_a$ | g m$^{-3}$ | Air density |
| $p_a$ | kPa | Air pressure |
| $e$ | kPa | Water vapor pressure |
| $R$ | kPa m$^3$ K$^{-1}$ mol$^{-1}$ | Universal gas constant |
| $R_d$ | kPa m$^3$ K$^{-1}$ g$^{-1}$ | Gas constant for dry air |
| $R_v$ | kPa m$^3$ K$^{-1}$ g$^{-1}$ | Gas constant for water vapor |



## C2  $\Delta CO_2$ estimates from IRGASON and $MGA^7$ measurements

To calculate the expected $CO_2$ concentration differences between the updraft and downdraft sample pairs collected in Zurich, the synchronized, despiked and to dry molar fractions converted 20 Hz $CO_2$ measurements were averaged over the periods where, according to the 20 Hz REA flags, air should have been collected to the up-/downdraft reservoir. For the IRGASON data, four sampling periods with low correlation between the $CO_2$ of the IRGASON and the $MGA^7$ (Pearson correlation coefficient < 0.5) due to a low $CO_2$ signal strength of the IRGASON (< 90 %) were discarded.

## C3  $\Delta CO_2$ with delayed collection of air

To estimate the potential bias and uncertainty caused by a certain time lag between the wanted and the actual collection of air, the $CO_2$ differences between REA sample pairs were calculated by shifting the 20 Hz REA flags forward in time and then averaging over the periods where the updraft or downdraft valves were open. These values were then compared to the $\Delta CO_2$ estimates calculated without time lag (Sect. C2). Table C2 shows the statistics of the differences in $\Delta CO_2$ with and without time lag for the 74 REA sampling periods in Zurich where $MGA^7$ data are available. The increasingly negative mean $\Delta CO_2$ differences show that $\Delta CO_2$ is systematically reduced when the collection of air is delayed. This is to be expected since in this case, air is collected when the vertical wind is within the deadband and $CO_2$ fluctuations are usually smaller than outside the deadband. With a time lag of 500 ms, as is expected in the Zurich setup, $\Delta CO_2$ is on average $0.04 \pm 0.06$ ppm smaller than in the ideal case without a delay. 0.06 ppm is therefore considered as the mean uncertainty due to a 500 ms delay in the collection of air. For 75 % of the REA runs, the change in $\Delta CO_2$ is less than 0.06 ppm, whereas the maximum simulated difference is 0.29 ppm. However, there is no systematic bias observed in the comparison between flasks and in situ (Fig. 6).

**Table C2.** Differences between $\Delta CO_2$ simulated with and without time lags between $w$ and $CO_2$ based on 20 Hz $CO_2$ measurements of the $MGA^7$. Given are the mean and standard deviation as well as the 0.25, 0.5, 0.75 and 1 quantiles of the Zurich REA runs (n = 74).

| Time lag [ms] | $\Delta CO_{2,lag}$ - $\Delta CO_2$ [ppm] | | $\|\Delta CO_{2,lag}$ - $\Delta CO_2\|$ [ppm] | | | |
|---|---|---|---|---|---|---|
| | Mean | Std | $Q_{0.25}$ | $Q_{0.5}$ | $Q_{0.75}$ | $Q_1$ |
| 100 | -0.002 | 0.010 | 0.002 | 0.003 | 0.007 | 0.041 |
| 200 | -0.009 | 0.023 | 0.004 | 0.008 | 0.017 | 0.100 |
| 300 | -0.018 | 0.034 | 0.003 | 0.014 | 0.029 | 0.156 |
| 400 | -0.028 | 0.047 | 0.008 | 0.017 | 0.042 | 0.226 |
| 500 | -0.040 | 0.061 | 0.011 | 0.025 | 0.058 | 0.285 |
| 600 | -0.051 | 0.075 | 0.016 | 0.033 | 0.069 | 0.356 |
| 700 | -0.063 | 0.088 | 0.019 | 0.040 | 0.085 | 0.426 |
| 800 | -0.075 | 0.101 | 0.021 | 0.045 | 0.099 | 0.494 |



## C4 $\Delta CO_2$ with incorrect rinse time

Uncertainty in the time an air parcel needs from the inlet to the buffers (about 2 s in Zurich) can cause sampling of unwanted air and loss of wanted air at the beginning and end of a REA run. If the rinse time were longer than the actual travel time by $\Delta t_r$ [s], the first $\Delta t_r$ seconds of sampled air would be lost, while air from $\Delta t_r$ seconds after the intended REA run end would be sampled. Similarly, if the rinse time were shorter than the actual travel time by $\Delta t_r$, unwanted air remaining in the intake lines would be sampled into the buffers, while the sample air from the last $\Delta t_r$ seconds in sampling mode would be lost. This was simulated by discarding and adding 20 Hz $CO_2$ measurements from the corresponding time periods and comparing the estimated concentration differences between updraft and downdraft samples ($\Delta CO_{2,\Delta t_r}$) with the (ideal) estimates where $\Delta t_r = 0$ s. The statistics from the 74 REA sampling periods in Zurich with MGA[7] data are shown in Table C3. The mean difference between $\Delta CO_2$ with and without error in the rinse time is 0.001 ppm, so well below the $CO_2$ measurement uncertainty of the flask samples. For individual REA runs, however, a 2 s over- or underestimation of the rinse time, can change $\Delta CO_2$ by up to 0.5 ppm. The standard deviation of about 0.01 ppm with $\Delta t_r = \pm 2$ s is considered as the $\Delta CO_2$ uncertainty of the Zurich samples due to the 2 s uncertainty in the rinse time. The fact that this standard deviation and the median and maximum differences are about twice as large when $\Delta t_r = \pm 4$ s shows, that this uncertainty contribution is site specific.

**Table C3.** $\Delta CO_2$ simulated from MGA[7] in situ measurements assuming different errors $\Delta t_r$ in the rinse time. Positive or negative $\Delta t_r$ indicate if the rinse time was overestimated or underestimated. Given are the mean and standard deviation as well as the 0.25, 0.5, 0.75 and 1 quantiles of the differences to the ideal case without $\Delta t_r = 0$ for the Zurich REA runs (n = 74).

| $\Delta t_r$ [s] | $\Delta CO_{2,\Delta t_r}$ - $\Delta CO_2$ [ppm] | | $\lvert\Delta CO_{2,\Delta t_r}$ - $\Delta CO_2\rvert$ [ppm] | | | |
| --- | --- | --- | --- | --- | --- | --- |
| | Mean | Std | $Q_{0.25}$ | $Q_{0.5}$ | $Q_{0.75}$ | $Q_1$ |
| -4 | 0.001 | 0.02 | 0.001 | 0.002 | 0.008 | 0.098 |
| -2 | 0.001 | 0.01 | 0 | 0.001 | 0.004 | 0.048 |
| 2 | 0.001 | 0.009 | 0.001 | 0.003 | 0.007 | 0.037 |
| 4 | 0.001 | 0.019 | 0.003 | 0.006 | 0.013 | 0.069 |

## C5 $\Delta CO_2$ with variable sampling rate

Under real sampling conditions, a certain variability of the flow rate and thus an inhomogeneous weighting of the concentration over time is to be expected, adding uncertainty to the $\Delta CO_2$ flask measurements. To estimate this uncertainty for each REA flask pair, the $CO_2$ concentration differences between updraft and downdraft samples were simulated from the high-frequency $CO_2$ measurements of the MGA[7] or, if not available, the IRGASON (compare Appendix C2), but this time with different weightings for each 20 Hz measurement. A total of 103 weighting functions were therefore calculated by interpolating the 103 flow rate time series (temporal resolution of 5 s) recorded by the two mass flow controllers in the loop systems during





the Zurich REA runs to the 20 Hz $CO_2$ in situ measurements of each REA run. This resulted in 103 $\Delta CO_2$ estimates for each REA run. The standard deviations of these estimates were then considered as the $\Delta CO_2$ uncertainty due to flow rate variability. The mean uncertainty of the 103 REA runs was $0.03 \pm 0.05$ ppm. As shown in Fig. C1, the magnitude depends on

the $CO_2$ variability of the ambient air. In the extreme case of a standard deviation of the ambient $CO_2$ of more than 50 ppm, the estimated uncertainty can be as high as 0.38 ppm. On the other hand, if $CO_2$ is approximately constant, the weighting does not matter. In the analysis of $\Delta CO_2$, e.g., in the comparison of flasks with in situ measurements (Sect. 5.3), it is therefore recommended to consider the uncertainty contributions for each REA sample individually. For the estimation of $\Delta ffCO_2$ on the other hand, the uncertainty due to flow rate variability is still negligible compared to the $^{14}C$ measurement uncertainty

(Appendix D).

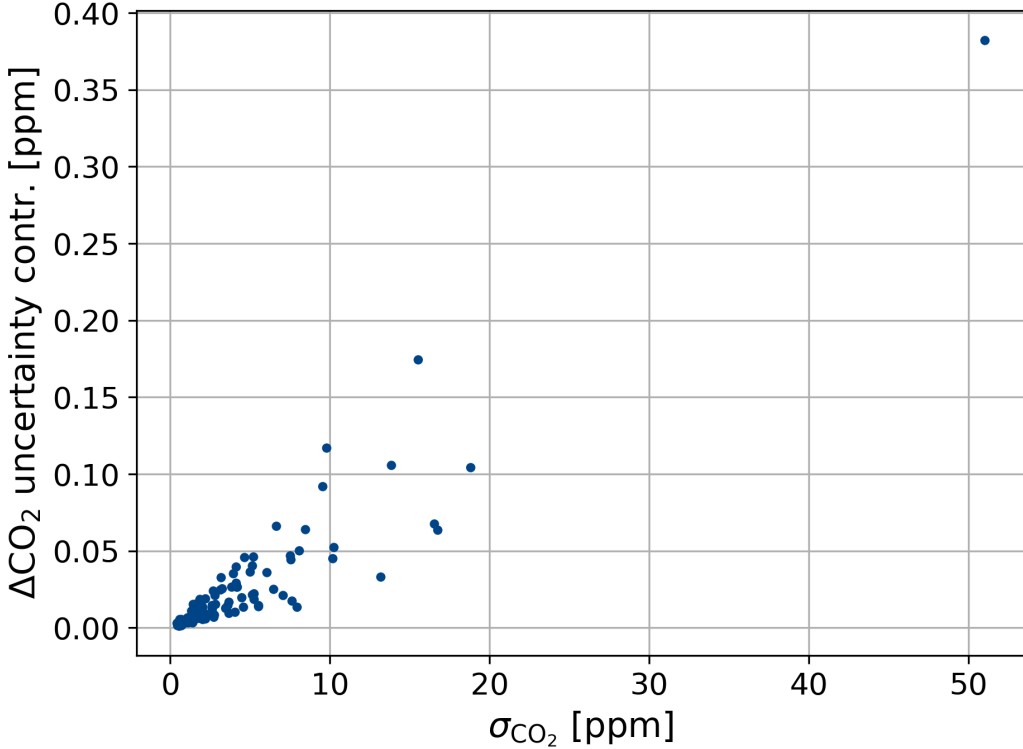

**Figure C1.** Estimated $\Delta CO_2$ uncertainty contribution due to a variability in the sampling flow rate as a function of the standard deviation $\sigma_{CO_2}$ of $CO_2$ of the ambient air during the sampling periods of the 103 REA Zurich samples.

**Appendix D: $\Delta ffCO_2$ uncertainties**

As mentioned in Sect. 3, $^{14}\Delta_{photo}$, $^{14}\Delta_{nf}$ and $\Delta c_{nf} = (c^{\uparrow}_{nf} - c^{\downarrow}_{nf})$, which are needed to calculate $\Delta ffCO_2$, are not known, but are estimated as given in Table 1. To illustrate the effects of each variable on the final $\Delta ffCO_2$ estimate, Fig. D1 shows the



differences between $\Delta ffCO_2$ calculated using the values in Table 1 and $\Delta ffCO_2$ when one of the variables is changed. For
the latter calculation, values over the maximum plausible range (based on $^{14}\Delta_{meas}$ and $\Delta c_{meas}$) and uncertainties close to mean
measurement uncertainties were chosen (2 ‰ for $^{14}\Delta_{nf}$ and $^{14}\Delta_{photo}$; 1 ppm for $\Delta c_{nf}$). Therefore, $\Delta(\Delta ffCO_2)$ plotted on the
y-axes represents the absolute error that would be made with the current assumptions if the "true" values were those given on
the x-axes.



**Figure D1.** Impact of different assumptions for $^{14}\Delta$-values of $CO_2$ differences between up and down flasks due to respiration and biofuels
(non-fossil, nf) and photosynthesis (photo), and the non-fossil signal $\Delta c_{nf} = c_{nf}^{\uparrow} - c_{nf}^{\downarrow}$ on estimates of $\Delta ffCO_2 = c_{ff}^{\uparrow} - c_{ff}^{\downarrow}$. For each REA
sample, the difference in $\Delta ffCO_2$ with respect to the reference settings used in this study is shown, where $^{14}\Delta_{nf} = 10 \pm 16$ ‰, $\Delta c_{nf} =$
$5 \pm 5$ ppm and $^{14}\Delta_{photo} = 0.5 \cdot (^{14}\Delta_{meas}^{\uparrow} + ^{14}\Delta_{meas}^{\downarrow}) \pm 10$ ‰. The light-blue ribbons indicate the $1\sigma$ uncertainty range due to measurement
uncertainties.



For all samples, higher $^{14}\Delta_{\mathrm{nf}}$ and $\Delta c_{\mathrm{nf}}$ lead to higher $\Delta\mathrm{ffCO_2}$ estimates. Consequently, with the assumptions of $^{14}\Delta_{\mathrm{nf}} = 9 \pm$
$16\,‰$ and $\Delta c_{\mathrm{nf}} = 5 \pm 5\,\mathrm{ppm}$, $\Delta\mathrm{ffCO_2}$ would be over-/underestimated if the actual values were smaller/larger. The magnitude
of the effect depends on the difference $^{14}\Delta_{\mathrm{nf}} - {}^{14}\Delta_{\mathrm{photo}}$ and is therefore largest for winter samples.

The effect of $^{14}\Delta_{\mathrm{photo}}$ is sample-specific because the reference values are different for each sample. However, for most
samples, $\Delta\mathrm{ffCO_2}$ decreases with increasing $^{14}\Delta_{\mathrm{photo}}$.

Varying the variables over the entire reasonable range, the differences can be as large as $2\,\mathrm{ppm}$ and thus in the same order of
magnitude as $\Delta\mathrm{ffCO_2}$ itself. Nevertheless, due to the large $^{14}\mathrm{CO_2}$ measurement uncertainty (compare light blue area in Fig.
D1), the deviations from the previous $\Delta\mathrm{ffCO_2}$ estimates are not significant.

The fact that the uncertainty of $\Delta\mathrm{ffCO_2}$ estimates is dominated by the $^{14}\mathrm{CO_2}$ measurement uncertainty can also be seen
in Fig. D2. It shows the contributions of each variable to the $\Delta\mathrm{ffCO_2}$ uncertainty in ppm for each REA sample pair. The
contributions of the $\Delta\mathrm{CO_2}$ uncertainty (including all aspects summarized in Table 3) and the assumptions on $^{14}\Delta_{\mathrm{nf}}$ and
$^{14}\Delta_{\mathrm{photo}}$ are summarized as "other assumptions and measurements" (black) and negligible for all samples. As discussed before,
the impact of $\Delta c_{\mathrm{nf}}$ (orange) is especially large for the winter samples, but still secondary compared to the $^{14}\Delta$ measurement
uncertainty (blue). The latter has been reduced from approximately $1.3\,\mathrm{ppm}$ to $1.0\,\mathrm{ppm}$, as the graphite targets have been
measured with a new accelerator mass spectrometer starting January 2023. Overall, the analyses show that the choice of
$^{14}\Delta_{\mathrm{photo}}$, $^{14}\Delta_{\mathrm{nf}}$ and $\Delta c_{\mathrm{nf}}$ has little effect on the $\Delta\mathrm{ffCO_2}$ estimate due to the current $\Delta^{14}\mathrm{C}$ measurement precision.





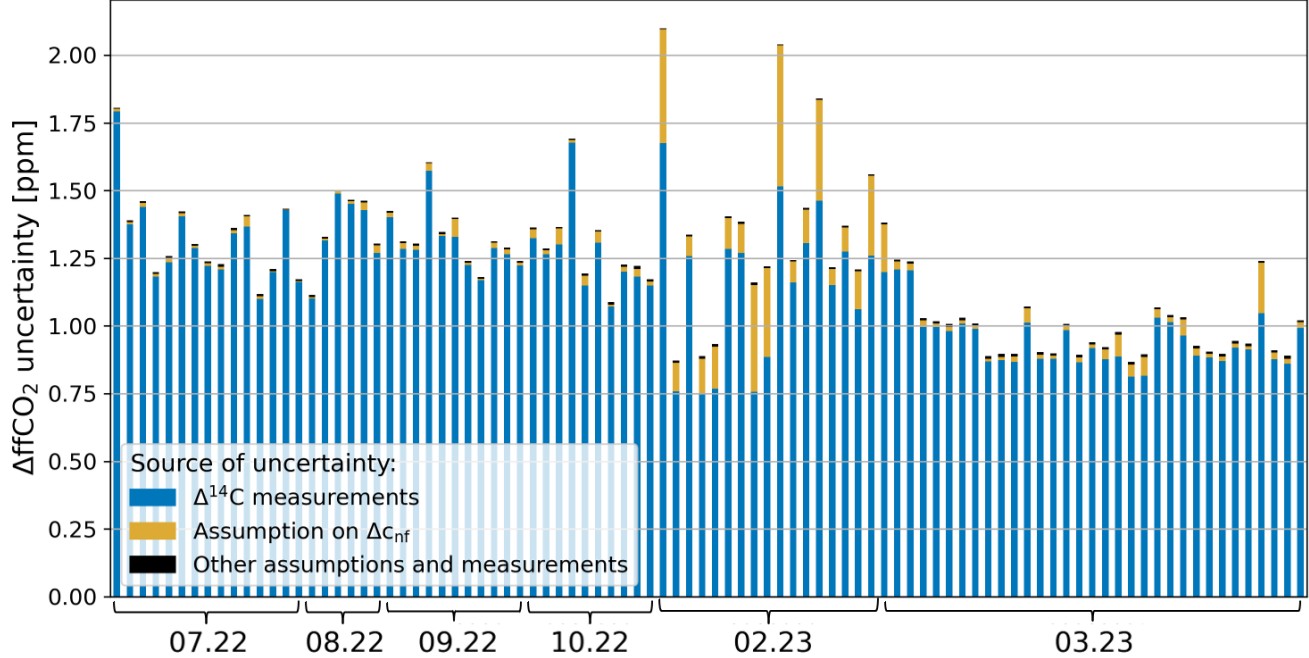

**Figure D2.** Contributions to the absolute uncertainty of $\Delta$ffCO$_2$ estimates (Eq. 8 with assumptions as given in Table 1) for each REA sample, collected between July 2022 and March 2023. $\Delta^{14}$C measurement uncertainties and an estimated non-fossil CO$_2$ difference between updraft and downdraft sample of $5 \pm 5$ ppm are the main contributors. The $\Delta$CO$_2$ uncertainties due to the sampling process and the laboratory analysis as well as the assumptions on the $\Delta^{14}$C signatures of photosynthesis, respiration and biofuels (here shown in black) are negligible.

*Author contributions.* IL and SH developed the idea of ffCO$_2$ REA measurements. VL, LB, RK and ME conducted initial proof-of-principle
tests, designed, built and tested the REA flask sampler. AC and JH performed numerical proof-of-concept simulations and developed the
logger code. JH, RH, AC and SS operated the IRGASON and MGA[7] in Zurich and provided data for sample selection and simulations of
flask concentration differences. AK, SH and IL collected and selected the REA samples during the Zurich campaign. PR exchanged the flasks
and maintained the REA sampler in Zurich. XG and JDC were responsible for the flask measurements at the FCL and the CRL, AJ and SP
for the respective data processing and quality control. AK prepared the manuscript with contributions from all co-authors.

*Competing interests.* The authors declare that they have no conflict of interest.

*Acknowledgements.* This project has received funding from the European Union's Horizon 2020 research and innovation program as part
of the Project "Pilot Applications in Urban Landscapes - Towards integrated city observatories for greenhouse gases" (PAUL) under grant
agreement No 101037319. Additional support was provided by internal funds and staff at the Universities of Heidelberg, Freiburg and the



Max Planck Institute for Biogeochemistry in Jena. We thank Lukas Emmenegger (EMPA, Switzerland) for negotiating and managing the

installation of the Zurich Hardau site, Roland Vogt (University of Basel, Switzerland) for installing the eddy covariance (IRGASON) system

in Zurich, Felix Baab (University of Freiburg, Germany) for constructing the REA inlets and tower valves. In addition, we gratefully thank

Steffen Knabe (ICOS FCL) and the whole staff of the ICOS Flask and Calibration Laboratory (FCL) and the Central Radiocarbon Laboratory

(CRL) for measuring the test and REA flasks. When comparing the REA flask concentrations with the IRGASON $CO_2$ measurements, we

appreciated the fruitful discussions with Ivan Bogoev (Campbell Scientific, Inc., USA, Logan).



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
