# Peer review of "A relaxed eddy accumulation flask sampling system for $^{14}$ C-based partitioning of fossil and non-fossil $CO_2$ fluxes"

_EGUsphere, 2024_

## Author Response (AR1)

Please find below the Referees' comments (blue font), followed by our detailed responses and the corresponding changes to the manuscript (black font). Unless otherwise stated, all page and line numbers refer to the originally submitted manuscript.

**Reply to Anonymous Referee #1**

Page 2 line 36. I agree that EC is the only method to directly measure fluxes. Given the discussion in the previous paragraph of inventory methods, it would be worthwhile to mention that there are multiple other atmospheric methods available besides EC. This is particularly relevant to this paper because while this paper presents the first time 14C has been used in REA, 14C measurements are widely used in other urban atmospheric methods, such as tracer ratios.

We agree that the paper, describing the novel application of REA to 14C measurements, benefits from a more comprehensive introduction that mentions other atmospheric methods using 14C to infer fluxes. We have therefore added a few sentences referring to three studies (Levin et al., 2003; Maier et al., 2024; Wu et al., 2022) that determined ffCO2 fluxes using 14C measurements. As the state-of-the-art laser-based 14C measurements and the lack of high-precision in situ measurements are closely linked, we combined the text with the changes done on page 2 lines 49-51 (see below).

Page 2 line 44. Human respiration is more typically ~5% of the total annual CO2 flux, and it depends not only on population density, but on emissions density.

We adopted the comment and adjusted the reference accordingly, as our previous references (Moriwaki and Kanda, 2004; Kellet et al., 2013) referred to individual examples where the contribution of human respiration to annual emissions was estimated to be more than 17 % and 8 %, respectively.

Page 2 lines 49-51. Suggest rephrasing this sentence, as it implies that in situ 14CO2 measurements are available, just not at fast-response. In fact, 14CO2 measurements can not yet be made in situ at all, except a few novel laser-based measurements that have not yet acheived the precision or method development to allow them to be used for atmospheric applications such as this.

We reformulated the sentence or section as suggested. In addition, we incorporated the changes from the above comment on page 2 line 36 here.

Page 13. Line 286. Suggest renumbering the 3 points as 5.1, 5.2, 5.3 to match the following section labels.

Done.

Page 14. Line 308. Here it says that the difference between two buffer fillings is  $0 \pm 52$  ppm. In the next sentence, the additional uncertainty is estimated at 0.15 ppm. I don't quite follow how this is calculated, and wonder whether the  $\pm 52$  ppm is a typo?

The numbers (± 52 ppm) are correct and the calculations are explained in Appendix B2. To avoid confusion, we omitted the number in the main text and refer the interested reader to Appendix B2.

Page 21. Line 470. Consider adding Turnbull et al 2015, Miller et al 2020 as additional references.

Turnbull JC, Sweeney C, Karion A, Newberger T, Lehman SJ, Tans PP, Davis KJ, Lauvaux T, Miles NL, Richardson SJ et al. 2015. Toward quantification and source sector identification of fossil fuel CO2 emissions from an urban area: Results from the INFLUX experiment. Journal of Geophysical Research: Atmospheres. 120.

Miller JB, Lehman SJ, Verhulst KR, Miller CE, Duren RM, Yadav V, Newman S, Sloop CD. 2020. Large and seasonally varying biospheric CO2 fluxes in the Los Angeles megacity revealed by atmospheric radiocarbon. Proceedings of the National Academy of Sciences of the United States of America.

Done.

Page 21-22 section 6.  $\Delta$ CO2 partitioning. The authors note that the  $\Delta$ 14C differences between up and downdrafts are small relative to the measurement uncertainty, resulting in many of the differences being indistinguishable from zero. There is some discussion in this section and also in appendix D about this, but the paper would benefit from expanding this discussion.

First, the actual  $\Delta 14C$  measurement precision acheived for these samples is not given, instead the uncertainty in calculated  $\Delta ffCO2$  is reported. It would be helpful to indicate what the  $\Delta 14C$  uncertainties are for these measurements. My estimate from the reported  $\Delta ffCO2$  uncertainties is that the  $\Delta 14C$  measurement uncertainties are around 2‰. Reducing these uncertainties would go a long way to improving the utility of the method. Several other labs are now reporting around 1.5‰ uncertainty on 14C measurements, and this modest improvement would make a significant difference to the fraction of usable measurements.

You are correct: the mean  $\Delta^{14}$ C uncertainty of all Zurich samples was 1.8 ‰, resulting in a mean  $\Delta$ ffCO2 uncertainty of 1.2 ppm. However, the uncertainty has already been reduced over the course of the campaign from 2.1 ± 0.3 ‰ to 1.6 ± 0.2 ‰, as the graphite targets have been measured on a new accelerator mass spectrometer (AMS) since January 2023. Currently, 1.6 ‰ is also the long-term reproducibility of the quality control targets at the AMS, and thus the minimum uncertainty of individual  $\Delta^{14}$ C measurements. For the REA measurements, however, even smaller uncertainties can be achieved by measuring updraft

and downdraft graphite targets in the same AMS magazine. Thus, we are currently at a mean  $\Delta^{14}C$  uncertainty of 1.1 ± 0.1 ‰ and a  $\Delta ffCO_2$  uncertainty of 0.7 ± 0.1 ppm. We could indeed significantly increase the fraction of usable measurements, as will be shown in a follow-up paper.

For the present paper we added the mean  $\Delta^{14}$ C uncertainties of the Zurich samples and refer to the improvements due to a new accelerator mass spectrometer on page 21 lines 481 - 484, in the conclusions on page 23 line 531, and in Appendix D page 34 lines 677-678 where the uncertainties are discussed in more detail.

Since the REA method doesn't appear to be sample size limited, one could also consider measuring multiple graphite targets to reduce the overall uncertainties. This would of course come at considerable cost in money and instrument time, but might be worth considering in the future.

In principle, this is an interesting idea. However, to effectively exchange the air in the flask with the sample air, the volume should be flushed 10 times (Levin et al., 2020). In addition, at least 6 I of air is required for laboratory analysis, and not all of the air in the buffers can be used because the pumps cannot evacuate the buffers in a reasonable time. Therefore, with the current setup, it is not possible to fill multiple flasks, i.e., obtain multiple graphite targets, from one REA sample. One could consider flushing two flasks in series with the same air, but in addition to a major reconstruction of the automated flask sampler, this would, as already mentioned, be associated with considerable costs in terms of money and instrument time. To share this knowledge, we added a comment on this on page 23, line 518.

Another option would be to make these REA measurements at EC sites that are closer to the surface and/or to sources, so that the  $\Delta 14C$  differences are larger. This would certainly be useful in demonstrating the technique, and could be a necessary constraint for the foreseeable future if  $\Delta 14C$  uncertainties cannot be beaten down further.

I don't suggest that the authors try to implement these things in this paper, but some discussion of these points would be helpful.

For a given surface flux and beta coefficient (e.g., given deadband width), the concentration difference between the updraft and downdraft samples depends mainly on the standard deviation of the vertical wind speed  $\sigma_w$ . Indeed, the latter is often smaller at lower heights, which would lead to an increased concentration difference (half  $\sigma_w \rightarrow$  double  $\Delta c$ ). However, at a lower measurement height, the flux footprint is much smaller, closer to the tower, and hence more heterogeneous in terms of sources. Consequently, the concentration differences are only larger if the sources are within this smaller footprint, and the measurements are only representative for this local area. Depending on the aim of a study, this may be fine, but for our purpose of urban measurements at the neighborhood to city scale, the EC and REA measurements should be made in the inertial sublayer (Feigenwinter et al., 2012). There,  $\sigma_w$ , and consequently  $\Delta c$ , are approximately constant with height. Therefore, we did not consider this to be a good solution for our purpose. We added a note to this effect on page 23, line 518.

Page 23. Lines 508-518. The same comments as above apply – indeed the signal-to-noise seems to be the main challenge.

We have expanded the discussion as explained above.

Page 23 line 522. Including CO and/or other species in these analyses would be very interesting. I wonder if incorporating CO measurements could also help resolve the signal-to-noise issues?

We agree that the analysis of CO/ffCO2 ratios will be interesting, since CO is often used as a tracer for ffCO2 emissions (see discussion above on other atmospheric measurement techniques). For individual REA measurements, however, this will not improve the signal-to-noise ratio. As for CO2, the signal-to-noise ratio will be largely dominated by the 14C-driven uncertainty of the ffCO2 estimate. We will examine and discuss the use of co-emitted species in detail in a follow-up paper.

**Reply to Anonymous Referee #2**

L13: At this point the mentioning of quality control flask pairs is quite cryptic to the reader. Depending on length limitations of the abstract, consider to either briefly explain what they are about, or reword like "112 flask pairs in total (103 for real-world fluxes and 9 for quality control purposes)".

We adapted the text as suggested: "103 REA up- and downdraft flask pairs for flux measurements and nine flask pairs for quality control purposes were selected".

L93: "with a dynamic deadband" sounds a bit arbitrary, maybe briefly refer to the standard deviation of w again (L72), or clearly define there that "dynamic deadband" in the rest of this manuscript will always refer to

In L93, we now refer once again to the standard deviation of the vertical wind velocity. Table 2 provides a clearer definition of "dynamic deadband", which is used from that point onwards.

L118: Given that the description is otherwise very detailed and inclusive for readers from outside the isotope community, wouldn't it be consistent to also briefly clarify the

normalization process, e.g. with another equation (I guess it would be the first one in sect. 2 of the cited source)?

We added the equation as suggested.

L126: abbreviating Delta^14 C by ^14 Delta looks a bit arbitrary and inconsistent. If it is rooted in common nomenclature of the isotope community, briefly mention it, if not consider a more consistent solution, like: Did I need the delta^14 C above in the first place? If yes, do I really need the abbreviation? If yes, can I at least keep the order between 14 and Delta consistent?

In our opinion, abbreviating  $\Delta^{14}C$  improves the readability of the equations and has also been done in other publications (e.g., Maier et al., 2023). In contrast to most publications, we originally wrote  $^{14}\Delta$  to avoid confusion with  $\Delta$  indicating a concentration difference between the updraft and downdraft samples. However, we agree that it is more consistent to keep the order between  $^{14}$  and  $\Delta$ . We have therefore changed it to  $\Delta^{14}$  and use this notation already in the definition in Eq. (3).

L142: "14Δphoto is best approximated by the current atmospheric signature 14Δmeas": Again, more explanation needed, it does not get clear why the current measured mixed signal should best approximate the pure signal of one of its components.

We added a note explaining that  $\Delta^{14}_{photo}$  equals the  $^{14}$ C signature of the photosynthesized atmospheric CO2, because the  $\Delta$ -notation accounts for mass-dependent fractionation. For our measurements of local CO2 fluxes close to the tall tower,  $\Delta^{14}_{photo}$  is therefore best approximated by the measured atmospheric signature  $\Delta^{14}_{meas}$ .

L146: "and set to 10": This statement only makes sense once the reader knows that measurement uncertainty is much smaller than that, maybe clarify by adding something, like at the end "which is well above the typical measurement uncertainty as will be shown in ....".

To enable the reader to better assess the stated  $\Delta^{14}_{photo}$  uncertainty of 10 ‰, we now refer to the mean measurement uncertainty of approximately 1.8 ‰ and to the more detailed analysis in Appendix D.

L172 ff. and Fig. 1: The order of introducing the buffer tanks before the loop system in the text, depicting the latter in Fig. 1 without any mentioning of a pump and then only clarifying much later in Fig. 2 is confusing. Fig. 1 as it is shown now would not work – readers must assume that the only pump in the system is the one mentioned in Fig. 1,

which would however leave unclear how air can leave the outflow (rather than being sucked in) and why all valves before the pump are closed at the moment of sampling.

We added the pumps to the loop systems in Fig. 1, and refer to them in L172ff (Fig. R1).

**Figure R1:** Schematic setup of the REA flask sampling system.  $P_a$  -  $P_c$  refers to the pumps that transfer the air from the inlets to flask sampler, and evacuate the buffers after sample transfer.  $p_1$  -  $p_4$  indicates pressure sensors. Blue components at the bottom of the diagram depict the general wiring for data transfer and communication between the data logger and the sampler computer. The green arrows and the filled / unfilled / hatched circles indicate the air flow and the position of the valves when sampling into buffer set 1.

L185: The photo reference is confusing, readers would first expect a reference to Fig. 3 which shows the inlets. Fig. 3 suggests that the 2 inlets were at the same height and horizontally displaced, while Fig. 1 suggests they are vertically displaced. Even if the figure is difficult to fix in that respect, it would be good to state the arrangement in the text (e.g. near L170).

We added a note on the horizontal displacement in L170, and in L185 we refer to both Fig.3 and Fig. A1.

L208: Out of the 3 valve state abbreviations NO, NC and CO mentioned in the Figure and caption, only one (NC) is referred to in the text. Establishing a better connection between figure, caption and text could help readers understand the system.

To help the reader understand the system, we adjusted the text in L208 and also mentioned the other valve state abbreviations, as suggested.

L221: tr is introduced as "an air parcel needs the time" above the equation and then redefined as rinse time two lines later. If it is the same, change the wording here, like e.g. "we refer to tr as rinse time in the following because it is exactly time the sampling needs to be artificially delayed to avoid sampling air from before the event".

As rinse time and "the time an air parcel needs" are the same, we changed the wording as suggested.

L238: maybe for discussion somewhere (not necessarily here): For which of the other gases would the REA system also be of interest due to a lack of fast analyzers? L240 "remaining": Does this only refer to the standard ICOS procedure or also to this manuscript, i.e. were the Zürich REA samples also analyzed for the other gases before extracting the CO2 for 14-C-analysis?

The Zurich REA samples were also analyzed for CO2, CO, CH4, N2O, SF6, and H2. We clarified this in L238. In addition, the  $\delta^{13}$ C,  $\delta^{18}$ O,  $\delta(O_2/N_2)$ , and  $\delta(Ar/N_2)$  concentrations were determined before the CO2 was extracted for the 14C analysis.

Of the gases mentioned, fast analyzers are available for direct eddy-covariance flux measurements of CO2, CO, CH4, and N2O. In Zurich, these gases were also continuously measured at 10 Hz with a closed-path multi-species infrared absorption gas analyser (MGA7) in parallel to the REA measurements. While the high-frequency EC measurements provide continuous and inexpensive flux data, the additional REA measurements can be used, for quality control purposes of the two methods, for example. They can also be used to analyse the beta coefficients and scalar similarity between different gases. For SF6, and H2,  $\delta^{13}$ C,  $\delta^{18}$ O,  $\delta$ (O2/N2), and  $\delta$ (Ar/N2), on the other hand, eddy covariance measurements are not possible to date due to a lack of fast response analyzers with sufficient precision. For these gases, the REA system could be used for flux estimation during the sampling periods. Multispecies analysis will be part of our future work.

The REA system can, of course, also be modified to measure other gases. In Zurich, the REA inlet lines and the logger program are currently used to measure VOC fluxes, for example.

L274: What does the lower flow rate at the MFC during rinse time imply? In hindsight, shouldn't then this be the flow to use in calculating rinse time? Or was the rest assumedly going through the outflow?

The flow at the MFC only affects the flow into the buffers (in sampling mode) or the lower part of the loop (in standby mode). The flow rate in the intake line remains approximately constant as a lower flow rate at the MFC results in a higher outflow/flow through the upper part of the loop (for very small flows through the MFC, the pump speed decreases due to increased pressure behind the pump, but in the case of 3.5 l/min instead of 4.7 l/min at the MFC, this effect is negligible). Therefore, it does not affect the rinse time. It only implies that the last few seconds of sampled air are slightly underrepresented. This effect was simulated and considered negligible (not shown here). However, it was still undesirable and avoidable. Based on this finding, the rinse time is now always set to the same value as during sampling. We added a short explanation in L274.

**L290: What is excluded by the words "only" and "focus", and why?**

With "only" and "focus" we wanted to emphasize that we analyze the uncertainty of the concentration difference between updraft and downdraft samples, rather than the uncertainty of the absolute concentration of individual samples. Since some non-idealities, e.g., memory and surface effects, affect updraft and downdraft samples similarly, we assume that the absolute concentrations have a larger uncertainty than the concentration differences. In addition, comparisons between flask and IRGASON or MGA7 measurements were possible only for concentration differences due to irregular calibration and the subsequent drift of the fast gas analyzers. Fortunately, only the concentration differences are important for later flux calculation (Eq. 1).

For clarification, we emphasize the difference between absolute concentrations and concentration differences already in L290 and not only in L291.

L312: Reference to appendix B3 missing?

Yes. added.

L322-323: It remains unclear how (and why?) only using the MGA here relates motivation-wise to comparing to both the IRGASON and MGA later as described near L390.

During the campaign, we compared the REA flask concentrations to both the IRGASON and the MGA7 measurements to check the performance of all three instruments and data processing. This was especially important as all of them were newly installed at the measurement site and/or novel instruments in general. For example, the effect of the intake lines of the REA and MGA7 was uncertain. The comparison also revealed that the default EC despiking algorithm was too strict for the urban area and erroneously identified point sources as instrument errors, leading to an underestimation of the measured concentration differences between the updraft and downdraft flasks. As we

found differences when comparing the flasks with the IRGASON and MGA7 measurements and as some measurement periods were only covered by one instrument, we show both comparisons in L390. For uncertainty analysis in L322, both data sets could have been used. We decided to use the MGA7, because these measurements agreed generally better with the flasks.

For clarification, we deleted "solely" in L323.

**L398: Clarify "1/t flow rate"**

A direct flask sample without usage of the buffers is taken by flushing air through a flask at a constant overpressure. To ensure that the flask concentration is as close as possible to the real mean concentration, the flow must be reduced over sampling time t according to 1/t (Levin et al., 2020). We added a short explanation in the text.

L432: How does the number 102 relate to the 103 from the abstract? If the difference results from the discarded measurements mentioned in the next sentence, why are they mentioned in plural?

A total of 103 flask sample pairs were analyzed for CO2, but only 102 measurements could be compared to IRGASON and/or MGA7 measurements because during one measurement period, both instruments were not operating well (i.e., from today's perspective, we would not have selected this sample for laboratory analysis). This was already stated in parentheses in the same sentence, but we added a few words to make it clearer.

L453: What "additional CO2 density output"? The text before suggests that raw CO2 density was available, and so were probably sonic temperature and H2O density, which is all that is needed to compute fast-response air temperature offline. An example for dry molar fraction is given in the appendix, but it can be done for any other measure of CO2 "concentration", and with even less slow-response data (basically only p), using H2O density from the IRGASON instead of q, only solving the equations gets a bit more complicated then.

The IRGASON measures  $CO_2$  absorption, which is then scaled with air temperature and pressure before being converted to  $CO_2$  density. Only the  $CO_2$  density is stored. Helbig et al. (2016) showed that the fast-response air temperature derived from sonic anemometer measurements should be used for this conversion. This output is referred to as "additional  $CO_2$  density output" and has been available since EC100 OS version 7.01. However, due to a lack of knowledge, our EC100 trigger was set to the mode that uses the slow-response temperature for the conversion. Since the raw absorption measurements are not stored, the  $CO_2$  density cannot be recalculated. Based on the

findings by Helbig et al. (2016), this could explain part of the bias. We tried to clarify this in L453.

460: 103 \*selected\* REA flask pairs seems to refer to the earlier mentioned fact that due to the costs of the 14CO2 analysis more flasks were taken but then carefully selected for the analysis. Then it is not understandable why subtracting so many more afterwards, except maybe the (unforeseeable) loss during graphitization. The 8 not analyzed could (as it looks now) as well be included in the selecting process from the start (thus lowering the number of 103), while for the 4 mentioned last it is unclear whether they were analyzed for 14CO2 and only discarded from data analysis afterwards. The way it is described now leaves unclear why the number 103 is important, as well as how objective the criteria for omitting the last 4 samples were.

The fact that eight sample pairs were analyzed for  $CO_2$  but not for  $^{14}CO_2$  was a consequence of our ongoing analysis and growing experience throughout this pilot application. For these eight samples, we decided only after the  $CO_2$  analysis at the ICOS Flask and Calibration Laboratory in Jena that, due to the small total  $CO_2$  differences and the expected  $ffCO_2$  uncertainties > 100%, measuring the  $^{14}C$  was not worth the costs. In retrospect, these samples could have been rejected from the beginning. However, the  $CO_2$  measurements are still valuable for assessing the quality of the REA system. Therefore, the measurements are included in Sect. 5. We extended the explanation in L461.

However, we agree that excluding further measurements requires a more detailed discussion. Since this is only important when the actual ffCO2 fluxes are calculated and analyzed quantitatively, we also show the four previously omitted concentration measurements in Fig. 7. In L484, we note that a detailed quality control of the existing dataset is important for flux estimation (see also L479/480). This will be considered in the follow-up paper.

L484 and L520: The paper stops somewhat abrupt, with everything on the table (at least according to how it is described in the methods section) to compute estimated fossil fuel fluxes but not presenting any. It is understandable that given the huge effort behind these measurements and the already long paper, the authors want to publish two separate papers about the methodological groundwork and the actual results. But then the title, abstract, parts of the introduction and methods section, frequent use of the word "partitioning" and presence of appendix D in this manuscript should be carefully reconsidered. For example, everything about estimating the deltas of other components seems to be needed here just for uncertainty estimations, and could more logically go into a later paper presenting the fluxes (together with their uncertainties).

We further emphasized that this paper only presents and analyzes the CO2 and 14CO2-based ffCO2 concentration differences of the REA flask pairs, whereas a subsequent calculation of the respective fluxes will follow in a separate paper. For this purpose, we ensured that we referred to the partitioning and assessment of the measurement uncertainties of the concentration differences (not to fluxes), e.g., in L3, L4, and L58.

Since Appendix D shows the uncertainties of the concentration differences, we believe this information is important for assessing the ffCO2 concentration differences presented in this paper.

L469-470 and 506-507: If data points to the right/below of the 1:1 line indicate respiratory and other non-fossil fuel signals and data points to the left/above photosynthesis, this means that any data point near the line could also result from simultaneous photosynthesis sinks and respiration/biofuel sources instead of from fossil-based CO2? This question is out of curiosity, and probably more relevant for a follow-up paper on the fluxes than for this one.

In the case of simultaneous photosynthetic uptake and respiration/biofuel sources only, the difference in the  $^{14}$ C signature between the updraft and downdraft samples would be much smaller than in the case of fossil fuel emissions. Consequently, the ffCO2 difference calculated from Eq. 9 would be zero (within the respective uncertainties), and the data point would lie on the x-axis in Fig. 7. If the flask measurements indicate that the ffCO2 difference is equal to the total CO2 difference (i.e., if the data point is on the 1:1 line), then photosynthetic uptake must have been compensated by non-fossil CO2 emissions. We now explain this more clearly in L468 ff.

L533-534: "water trap operated in reversed order" = (custom) dewpoint generator? Were the 12°C a setting or an uncontrollable result of the device's setup? How does the humidification avoid surface effects?

As the setup with the water trap operated in reversed order functioned as a dew point generator, we adapted the wording in L533 and L544. The dew point was determined by the temperature of the water used to humidify the gas. This temperature could be controlled by the temperature of a silica oil bath and was approximately 12 ± 1 °C. As the system is usually operated with (humid) ambient air, and the strength of surface effects, i.e., the adsorption and desorption of CO2 molecules by a surface, depends on the humidity/water availability (e.g., Zhao et al, 2020), we tried to minimize potential biases from increased/reduced surface effects by humidifying the test gas. As the description in L533-534 may have been imprecise, we have adapted it accordingly.

**References:**

- Feigenwinter, C., Vogt, R., and Christen, A.: Eddy Covariance Measurements Over Urban Areas, in: Eddy Covariance, edited by Aubinet, M., Vesala, T., and Papale, D., pp. 377–398, Springer, Dordrecht, https://doi.org/10.1007/978-94-007-2351-116, 2012.
- Helbig, M., Wischnewski, K., Gosselin, G. H., Biraud, S. C., Bogoev, I., Chan, W. S., Euskirchen, E. S., Glenn, A. J., Marsh, P. M., Quinton, W. L., and Sonnentag, O.: Addressing a systematic bias in carbon dioxide flux measurements with the EC150 and the IRGASON open-path gas analyzers, Agr. Forest Meteorol., 228-229, 349–359, https://doi.org/10.1016/j.agrformet.2016.07.018, 2016.

- Kellett, R., Christen, A., Coops, N. C., van der Laan, M., Crawford, B., Tooke, T. R., and Olchovski, I.: A systems approach to carbon cycling and emissions modeling at an urban neighborhood scale, Landscape Urban Plan., 110, 48–58, https://doi.org/10.1016/j.landurbplan.2012.10.002, 2013.
- Levin, I., Kromer, B., Schmidt, M., and Sartorius, H.: A novel approach for independent budgeting of fossil fuel CO2 over Europe by 14CO2 observations, Geophys. Res. Lett., 30, https://doi.org/10.1029/2003GL018477, 2003.
- Levin, I., Karstens, U., Eritt, M., Maier, F., Arnold, S., Rzesanke, D., Hammer, S., Ramonet, M., Vítková, G., Conil, S., Heliasz, M., Kubistin, D., and Lindauer, M.: A dedicated flask sampling strategy developed for Integrated Carbon Observation System (ICOS) stations based on CO2 and CO measurements and Stochastic Time-Inverted Lagrangian Transport (STILT) footprint modelling, Atmos. Chem. Phys., 20, 11 161–11 180, https://doi.org/10.5194/acp-20-11161-2020, 2020.
- Maier, F., Levin, I., Conil, S., Gachkivskyi, M., van der Denier Gon, H., and Hammer, S.: Uncertainty in continuous  $\Delta$ CO-based  $\Delta$ ffCO2 estimates derived from 14C flask and bottom-up  $\Delta$ CO /  $\Delta$ ffCO2 ratios, Atmos. Chem. Phys., 24, 8205–8223, https://doi.org/10.5194/acp-24-8205-2024, 2024.
- Moriwaki, R. and Kanda, M.: Seasonal and Diurnal Fluxes of Radiation, Heat, Water Vapor, and Carbon Dioxide over a Suburban Area, J. Appl. Meteorol., 43, 1700–1710, https://doi.org/10.1175/JAM2153.1, 2004.
- Wu, K., Davis, K. J., Miles, N. L., Richardson, S. J., Lauvaux, T., Sarmiento, D. P., Balashov, N. V., Keller, K., Turnbull, J., Gurney, K. R., Liang, J., and Roest, G.: Source decomposition of eddy-covariance CO2 flux measurements for evaluating a high-resolution urban CO2 emissions inventory, Environ. Res. Lett., 17, 074 035, https://doi.org/10.1088/1748-9326/ac7c29, 2022.
- Zhao, J., Deng, S., Zhao, L., Yuan, X., Du, Z., Li, S., Chen, L., and Wu, K.: Understanding the effect of H2O on CO2 adsorption capture: mechanism explanation, quantitative approach and application, Sustainable Energy Fuels, 4, 5970-5986, https://doi.org/10.1039/D0SE01179G, 2020.